

# The effect of organic nucleation on the indirect radiative forcing with a semi-explicit chemical mechanism for highly oxygenated organic molecules (HOMs)

Xinyue Shao[1,2], Minghuai Wang[1,2], Xinyi Dong[1,2,3], Yaman Liu[1,4], Stephen R. Arnold[5], Leighton A. Regayre[5,6,7], Duseong S. Jo[8], Wenxiang Shen[1,2], Hao Wang[1,2], Man Yue[1,4], Jingyi Wang[1,2], Wenxin Zhang[1,2], and Ken S. Carslaw[5]

[1]School of Atmospheric Science, Nanjing University, Nanjing, 210023, China
[2]Joint International Research Laboratory of Atmospheric and Earth System Sciences & Institute for Climate and Global Change Research, Nanjing University, Nanjing, 210023, China
[3]Frontiers Science Center for Critical Earth Material Cycling, Nanjing University, Nanjing, China
[4]Zhejiang Institute of Meteorological Sciences, Hangzhou, 310008, China
[5]School of Earth and Environment, University of Leeds, Leeds, LS2 9JT, UK
[6]Met Office Hadley Centre, Exeter, Fitzroy Road, Exeter, Devon, EX1 3PB, UK
[7]Centre for Environmental Modelling and Computation, School of Earth and Environment, University of Leeds, Leeds, LS2 9JT, UK
[8]Department of Earth Science Education, Seoul National University, Seoul, 08826, South Korea

*Correspondence to*: Minghuai Wang (minghuai.wang@nju.edu.cn), Xinyi Dong (dongxy@nju.edu.cn)

**Abstract.** Highly oxygenated organic molecules (HOMs) can significantly contribute to new particle formation (NPF). HOMs-derived NPF in preindustrial (PI) environments provides the baseline for calculating radiative forcing, yet global model studies examining this are lacking. Here, we use a global climate model with a semi-explicit HOMs chemistry and the associated nucleation scheme to systematically quantify the effect of HOMs-derived NPF on CCN formation and effective radiative forcing due to aerosol–cloud interactions ($ERF_{aci}$). The model shows better agreement with measured cloud condensation nuclei (CCN) numbers after including organic NPF mechanisms. Aerosols generated from organic NPF nearly double the globally averaged CCN burden in PI (39%) compared to PD (18%) experiments. This weakens the $ERF_{aci}$ by 0.4 W m$^{-2}$, corresponding to a 16% reduction, with most of this reduction occurring in tropical regions where the pure organic nucleation rate shows larger value in PI atmosphere. Unlike the findings of Gordon et al. (2016), the reduction is mainly driven by a greater enhancement of the sub-20 nm growth rate (GR) in the PI atmosphere compared to PD instead of the ~1 nm nucleation rate ($j_{1.7nm}$). The greater enhancement of GR is due to higher HOM concentrations in the PI atmosphere, while the greater $j_{1.7nm}$ in PD environment results from higher sulfuric acid concentrations, leading to higher heteromolecular nucleation rates involving sulfuric acid and organics. The significant reduction underscores the critical role of biogenic NPF in CCN formation, particularly in the PI climate when cloud droplet concentrations and albedo are more sensitive to aerosol changes.



## 1 Introduction

Atmospheric aerosols can affect climate indirectly by acting as cloud condensation nuclei (CCN), which modify cloud properties and precipitation (Rosenfeld and Lensky, 1998; Rosenfeld and Woodley, 2000; Twomey, 1977; Albrecht, 1989). The effective radiative forcing due to aerosol–cloud interactions (ERF$_{aci}$) remains one of the largest uncertainties in interpreting climate change over the past century and projecting future changes (Watson-Parris and Smith, 2022; Peace et al., 2020). Atmospheric new particle formation (NPF) is the largest source of atmospheric aerosol number concentrations (Gordon et al.,

2017; Lee et al., 2019) and is thought to contribute up to half of the global cloud condensation nuclei (CCN) number (Spracklen et al., 2008; Merikanto et al., 2009; Williamson et al., 2019). Global climate model simulations indicate that aerosol ERF$_{aci}$ is sensitive to parameterizations of NPF processes (Wang and Penner, 2009; Kazil et al., 2010; Gordon et al., 2017; Zhu et al., 2019; Yu et al., 2012).

Recent studies have highlighted the significant role of monoterpene-derived highly oxygenated organic molecules (HOMs) in NPF processes and their potential impact on regulating CCN concentrations, even in the absence of sulfuric acid. Ehn et al. (2014) found significant contributions of HOMs to the growth of particles ranging from 5 to 50 nm in diameter in boreal forests. Jokinen et al. (2015) showed that monoterpene-derived HOMs facilitate new particle formation and growth in continental regions, especially under conditions of high supersaturation, favorably affecting the concentration of CCN using chamber

experiments. Using the regional model WRF-Chem, Zhao et al. (2020) showed that in the Amazon region biogenic HOMs predominantly lead to the formation of new particles at altitudes of 13 km, in a location minimally influenced by human activities, thereby making a significant contribution to CCN formation (Zhao et al., 2022). Similarly, Wang et al. (2023) analyzed the sources of aerosols in the Amazonian boundary layer using the HadGEM3 climate model incorporating the biogenic nucleation mechanism along with its precursor gas (HOMs) from the parameterization in Gordon et al. (2016).


Although HOMs are important for NPF due to their low volatility, their chemical formation pathways remain uncertain, and are they treated in various simplified ways in models. Gordon et al. (2016) simulated monoterpene-derived HOMs formation using a fixed yield of HOMs from first-stage monoterpene oxidation products. Zhu et al. (2019) added some explicit chemical mechanisms for HOMs, but they did not consider autoxidation and used a less stringent definition of HOMs than recommended

in Bianchi et al. (2019). Also, they did not account for organic nucleating species oxidized from isoprene. Therefore, the contribution of accretion products (ACCs) generated from cross-reactions of isoprene- and monoterpene-derived radicals was significantly underestimated. Roldin et al. (2019) and Weber et al. (2020) employed a more explicit reaction mechanisms to treat the generation of HOMs through autoxidation and cross-reactions of α-pinene oxidation products, but neither applied these chemical mechanisms on a global scale. Xu et al. (2022) summarized various chemical mechanisms of HOMs, including

monoterpene-derived peroxy radical (MT-RO$_2$) unimolecular autoxidation and self- and cross-reactions with other RO$_2$ species in the GEOS-Chem global model but did not consider their role in the NPF process.



HOM-driven NPF is likely to be particularly important in the pristine preindustrial (PI) atmosphere, where concentrations of sulfuric acid and ammonia were much lower. Simulations of the pre-industrial atmosphere form the baseline for calculations of anthropogenic radiative forcing in global models (Carslaw et al., 2013). Using global model simulations, Gordon et al. (2016) showed that new particles formed from monoterpene-derived HOMs could increase CCN concentrations in the PI environment by 20% to 100%. This increase led to a 27% reduction in negative radiative forcing due to changes in cloud albedo since 1750, decreasing by between -0.28 W m$^{-2}$ and -0.06 W m$^{-2}$. Similarly, Zhu et al. (2019), utilizing simulations with the Community Earth System Model (CESM), found that new particles formed from monoterpene-derived HOMs have reduced the direct plus indirect radiative forcing of aerosols by 12.5% since the Industrial Revolution. However, they did not use explicit chemical processes to represent HOMs chemistry, potentially leading to inaccuracies in the PI and present-day (PD) simulations for radiative forcing calculations due to anthropogenic emissions. The uncertainty in this baseline is one of the largest components of the overall uncertainty in aerosol radiative forcing (Seinfeld et al., 2016; Carslaw et al., 2013).

Considering the unequivocal evidence for the role of biogenic organics in producing atmospheric particles, we have recently incorporated a state-of-the-art representation of HOMs from various chamber experiments (Xu et al., 2022b) and update the nucleation scheme involving HOMs, ACCs, $H_2SO_4$, $NH_3$, and ions from previously published CLOUD chamber experiments in a global chemistry-climate model (Shao et al., 2024). Additionally, Shao et al. (2024) account for organics condensing on newly formed sub-20 nm particles. The updated model demonstrates significant improvements in simulating NPF events and aerosol number concentrations, showing better agreement with measurements (Shao et al., 2024). Here, we seek to estimate the change in ERF$_{aci}$ resulting from the inclusion of organic particle formation based on this model, and highlight the key processes driving this change.

The model and field measurements used in this study are documented in Section 2. Section 3 evaluates CCN number concentrations in the updated model. Section 4 quantifies the contributions of organic NPF to CCN number globally in both present day (PD) and PI environments. The change in effective radiative forcing due to aerosol–cloud interactions (ERF$_{aci}$) associated with organic NPF processes is also calculated. Results are summarized and discussed in Section 5.

## 2 Data and methods

### 2.1 Model configuration

In this study, we examine the impact of organic NPF on atmospheric aerosols and the Earth's radiative balance using the atmospheric module of the Community Earth System Model (CESM) version 2.1.0, specifically the Community Atmosphere Model version 6, which is enhanced with extensive tropospheric and stratospheric chemistry (CAM6-Chem) (Emmons et al., 2020). The model uses the MOZART-TS2 gas-phase chemistry scheme (Schwantes et al., 2020) and employs a four-mode



version of the Modal Aerosol Module (MAM4) (Liu et al., 2016). The default configuration of CAM6-Chem includes binary homogeneous nucleation of $H_2SO_4$–$H_2O$ (Vehkamaki et al., 2002) and ternary homogeneous nucleation of $H_2SO_4$–$NH_3$–$H_2O$ (Merikanto et al., 2007). Additionally, within the boundary layer, the model includes the empirical nucleation mechanism (Kulmala et al., 2006; Sihto et al., 2006).


Our previous study (Shao et al., 2024) incorporated the representation of HOMs chemistry from Xu et al. (2022a) (including monoterpene derived peroxy radical (MT-$RO_2$) unimolecular autoxidation and self- and cross-reactions with other $RO_2$ species). In total, 24 reactions in CAM6-Chem were modified and 96 reactions were added to more explicitly simulated HOMs chemistry (Section S1). Shao et al. (2024) also updated inorganic nucleation rates involving $H_2SO_4$ and $NH_3$ as well as ion-induced pathways based on the CLOUD chamber experiments (Dunne et al., 2016), replacing the default scheme based on $H_2SO_4$ and $NH_3$ (Vehkamaki et al., 2002; Merikanto et al., 2007). The organic nucleation scheme was also added in CAM6-Chem, including heteromolecular nucleation of sulfuric acid and organics ($J_{SA-Org}$) (Riccobono et al., 2014), neutral pure organic nucleation ($J_{Org,n}$) and ion-induced pure organic nucleation ($J_{Org,i}$) (Kirkby et al., 2016). Organic vapor condensation on newly formed particles was also added in our updated model (Eq. (12) in Shao et al. (2024)).


All the above mentioned updated nucleation rate and sub-20 nm particle growth rates have already been evaluated in our previous study (Shao et al., 2024), in better agreement with observations at numerous sites. Therefore, the performance of NPF event frequency and N10 (number concentrations for particles with diameters larger than 10 nm) also shows reasonable agreement with measurements (Shao et al., 2024).


**2.2 Case setting**

We calculated the $ERF_{aci}$ between preindustrial (PI) and present-day (PD) using the methodology from Ghan (2013). The effect of biogenic organic NPF on the magnitude of $ERF_{aci}$ was calculated by comparing simulations with (using Eq. (2)-(8) in Shao et al. (2024), named "Inorg_Org" in table 1) and without (only using Eq. (6)-(8) in Shao et al. (2024), named "Inorg" in table 1) biogenic NPF mechanisms. The prefixes "PD" and "PI" in each test name represent emissions scenarios appropriate to present day and preindustrial (Table 1).

Ten-year simulations were performed with 0.9° × 1.25° spatial resolution and a vertical resolution extending up to approximately 40 km across 32 layers (Emmons et al., 2020) for both present-day (PD) and preindustrial (PI) atmospheres with an additional one-year spin-up period (Table 1). Sea surface temperature and sea-ice extents are prescribed to climatological values for the year 2000 in both PD and PI cases. Anthropogenic and monthly biomass burning emissions are provided by the Community Emission Data System (CEDS v2017-05-18) (Hoesly et al., 2018) and the historical global biomass burning emissions inventory (van Marle et al., 2017) developed for CMIP6. For PD simulation, emissions after 2014 follow the SSP585 scenario, based on the Shared Socioeconomic Pathway 5 (SSP5) (O'Neill et al., 2017). Biogenic emissions



are dynamically simulated using the Model of Emissions of Gases and Aerosol from Nature version 2.1 (MEGAN2.1) (Guenther et al., 2012). The Multi-resolution Emission Inventory for China (MEIC) (http://www.meicmodel.org) (Li et al., 2017; Yue et al., 2023) was used to replace the CMIP6 emission inventory for China, as CMIP6 underestimates the reduction of $SO_2$ emissions after 2007.

In order to compare simulated CCN with measurements, several short-term simulations were performed, in which meteorological fields (temperature and wind profiles, surface pressure, surface stress, surface heat and moisture fluxes) were nudged toward Modern-Era Retrospective analysis for Research and Applications (MERRA2) reanalysis (Kooperman et al., 2012). The simulation period corresponded to the time of measurements (Table 2), with an additional month for spin-up.

**Table 1.** Configurations of CESM2.1.0 Experiments (long-term simulation)

| Test Name | Simulation period | Spin-up | Updated inorganic nucleation | HOMs chemistry | Organic nucleation and growth |
|---|---|---|---|---|---|
| Inorg | corresponds to the measurements in Table 2 | one month | ✓ | ✓ | × |
| Inorg_Org | | | ✓ | ✓ | ✓ |
| PD_Inorg | 2008.1-2017.12 | one year (2007.1-2007.12) | ✓ | ✓ | × |
| PD_Inorg_Org | | | ✓ | ✓ | ✓ |
| PI_Inorg | 1851.1-1860.12 | one year (1850.1-1850.12) | ✓ | ✓ | × |
| PI_Inorg_Org | | | ✓ | ✓ | ✓ |

**2.3 Observation data**

In our previous study (Shao et al., 2024), we evaluated NPF-related variables, including nucleation rate, growth rate, NPF
frequency, condensation sink, and aerosol number concentration in the updated model. Here, we focus specifically on evaluating the CCN number concentration in both Inorg_Org and Inorg models. The CCN number concentration is crucial because it influences the degree of cooling of the Earth's surface through the aerosol-cloud interaction, specifically by influencing cloud albedo and cloud lifetime (Twomey, 1977; Albrecht, 1989).

The observational data of CCN number concentrations used in this study were obtained from ships, stations, and aircraft at various locations (see Table 2) (Jefferson, 2010; Uin et al., 2019; Bodhaine, 1983; Wang et al., 2022; Wood et al., 2015; Zheng



et al., 2020) (all the data are available for download at http://www.archive.arm.gov/discovery/#v/results/). All data were processed within the Global Aerosol Synthesis and Science Project (GASSP) (Reddington et al., 2017). The CCN number measurements exhibit a very high temporal resolution (<1 minute). However, the model output's physical time step is only half an hour, making it impossible to precisely match the observation data with the model output. Consequently, we selected CCN number concentrations at supersaturations (ss) of 0.1%, 0.2%, 0.5%, and 1% from the observational data to represent different atmospheric supersaturations. These values were then compared with the model's monthly average output at the same supersaturation levels for the corresponding time period (Fig. 1).

**Table 2.** Field measurements used in this study

| Station | Type | Altitude (m asl) | Latitude | Longitude | Time |
|---|---|---|---|---|---|
| Barrow, Alaska, USA | Marine | 8.0 | 71.32° N | 156.61° W | 2008/01-2013/01 |
| Eastern North Atlantic | Marine | 26.0 | 39.09° N | 28.03° W | 2014/11-2015-04 |
| Graciosa Island | Marine | 26.0 | 39.09° N | 28.03° W | 2009/05-2010/11 |
| Manacapuru, Brazil | Urban | 50.0 | -3.21° S | 60.60° W | 2014/02-2015/05 |
| Nainital, India | Mountain | 1936.0 | 29.36° N | 79.46° E | 2011/06-2012/03 |
| Shouxian, China | Rural | 22.7 | 32.56° N | 116.78° E | 2008/05-2008/10 |
| Steamboat Springs, USA | Mountain | 2440.0 | 40.45° N | 106.79° W | 2010/10-2011/04 |
| Chilbolton, UK | Rural | 80.0 | 51.15° N | 1.44° W | 2009/02-03 |

**3 Evaluation of simulated CCN concentrations**

CCN number concentrations in the Inorg_Org model at 0.1%, 0.2%, 0.5%, and 1% supersaturation (ss) show better agreement with measurements from various locations compared to the Inorg model (Fig. 1). The underestimation of CCN numbers in the Inorg simulation is alleviated by incorporating organic-related NPF, especially over rural and mountainous regions such as Steamboat Springs, Shouxian, and Nainital (Fig. 1), where both nucleation and initial growth rates are dominated by biogenic pathways. However, there remains a slight underestimation of CCN number in Shouxian at all supersaturation levels. This is likely due to the neglect of anthropogenic-derived HOMs to nucleation and growth, which are key NPF mechanisms in rural regions of China. The increase in CCN number due to the addition of organic NPF mechanisms is simulated not only in the locations listed in Table 1 but also on a global scale (see Fig. 7). "In urban regions of Brazil (Manacapuru), the overestimation of CCN numbers at 0.5% and 1% ss in Inorg is exacerbated (Fig. 1). Apart from urban regions like Manacapuru, Brazil, the overestimation also increases over oceanic regions such as Barrow and Graciosa. These overestimations in CCN numbers in the Inorg model are likely related to the overestimation of $H_2SO_4$ concentration in CAM6-Chem (Shao et al., 2024). Overall,



the normalized mean bias (NMB) of CCN numbers at different supersaturation levels decreases from -35% (Inorg) to -24% (Inorg_Org), indicating that the Inorg_Org model provides a more accurate representation of organic contributions for further quantification in Section 3.

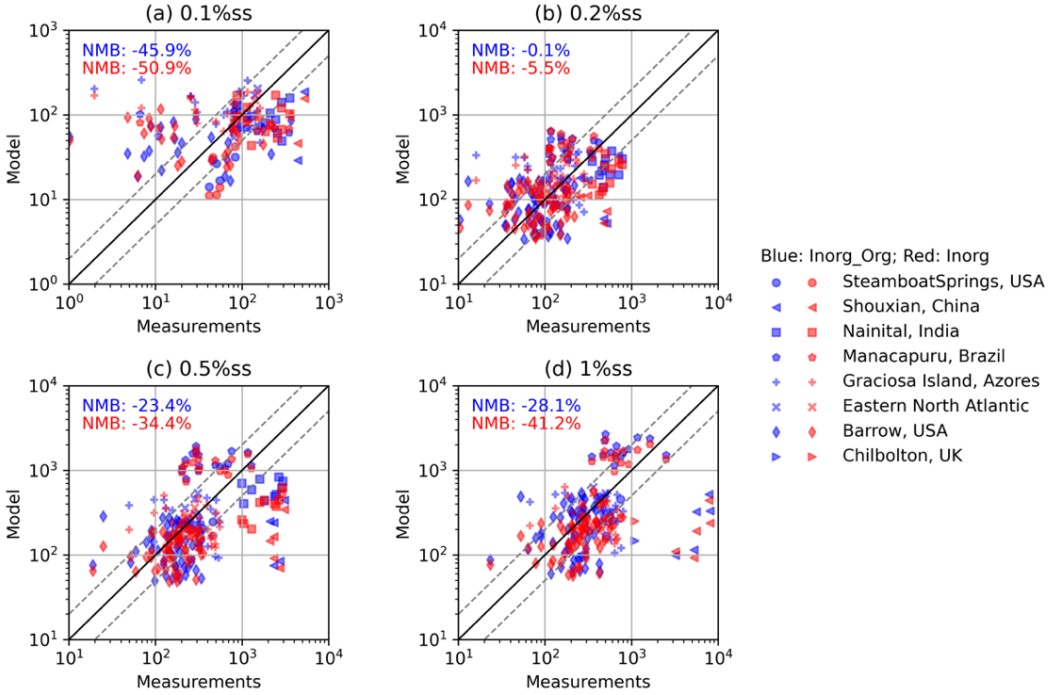

**Figure 1.** Comparison of simulated (monthly mean) and measured CCN number (median value) concentration (unit: cm⁻³) at **(a)** 0.1% supersaturation (ss), **(b)** 0.2% ss, **(c)** 0.5%ss, and **(d)** 1% ss in Inorg_Org (blue symbols) and Inorg (red symbols). Information regarding the measurement sites is summarized in **Table 2**. Normalized mean bias (NMB) values are shown on the top left of each figure.

## 4 Results

### 4.1 Change in CCN and cloud droplet number concentrations

The inclusion of organic NPF results in a greater increase in the CCN burden in the PI experiment (~39%) compared to the PD experiment (~18%) (Fig. 2). The spatial pattern of change in CCN burden (Fig. 2) is consistent with the change in aerosol burden in the Aitken and accumulation mode (Figs. S11 and S12) but affects much wider areas because of the slower removal rate of larger aerosol particles. On the western side of the Amazon basin, the highest rise (>50%) in both the PD and PI experiments is caused by high CCN burden transported from the Amazon basin. Such a significant change in CCN production across the unpolluted PI atmosphere is particularly important for global climate because cloud droplet number concentrations (CDNC) are sensitive to CCN changes. Therefore, CDNC at the top of low clouds in Inorg_Org rise by 12% in the PI experiments but only 7% in the PD experiments compared to Inorg (Fig. 3).



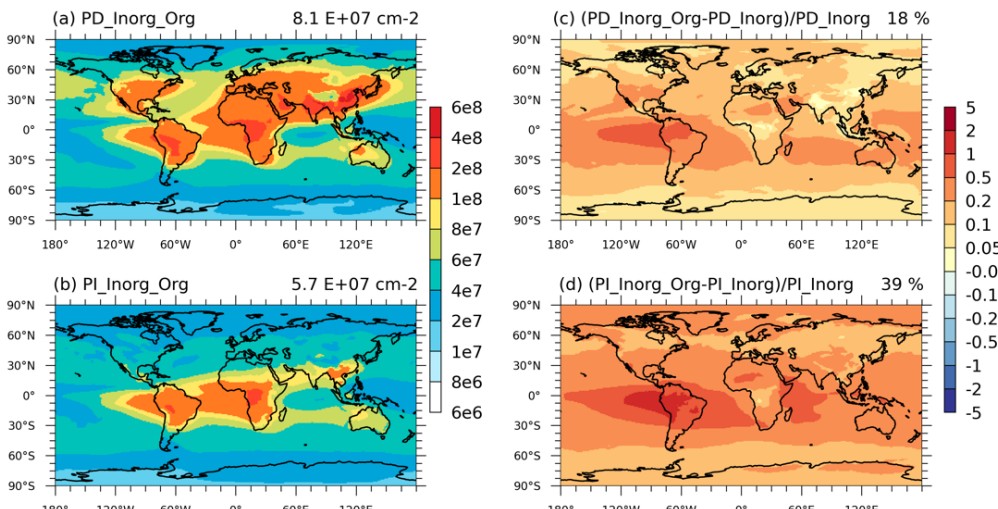

**Figure 2.** Spatial distribution of the simulated vertically-integrated CCN at 0.2% supersaturation (ss) in **(a)** PD_Inorg_Org and **(b)** PI_Inorg_Org (unit: cm⁻²). The relative change after adding organic NPF is shown in PD and PI environments are shown in **(c)** and **(d)**. Global mean values are shown on the top right of each figure.

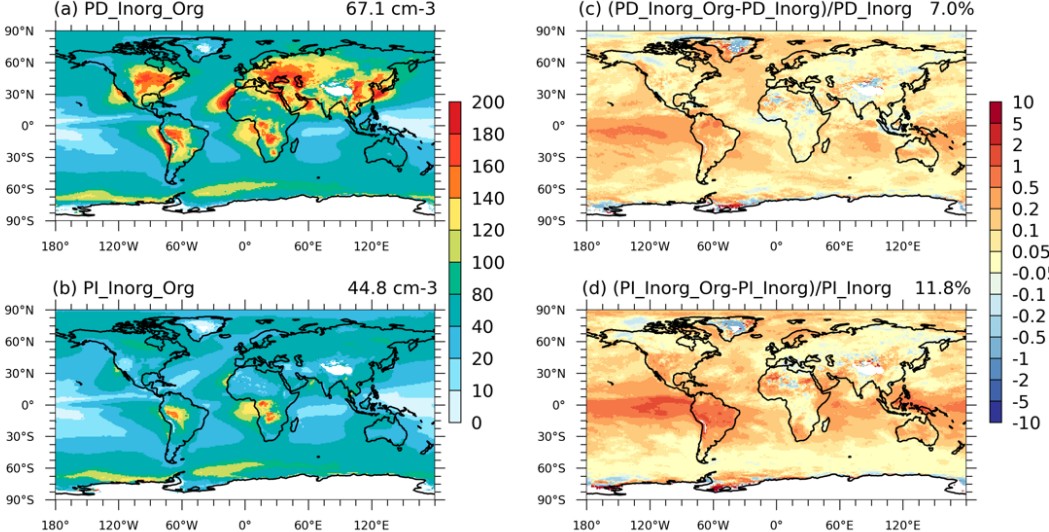

**Figure 3.** Spatial distribution of the simulated vertically-integrated cloud droplet number concentration (CDNC) at the top of low clouds in **(a)** PD_Inorg_Org and **(b)** PI_Inorg_Org (unit: cm⁻³). The relative change after adding organic NPF is shown in PD and PI environments are shown in **(c)** and **(d)**. Global mean values are shown on the top right of each figure.

In both PD and PI experiments, the largest increase in CCN burden (>20% rise in Inorg_Org compared to Inorg) is simulated in the tropical regions (Amazon, central Africa, and Southeast Asia) (Fig. 2). This is attributed to the highest biogenic emissions

(Fig. S3) which lead to the greatest increases in both nucleation and growth rates in Inorg_Org (Fig. 4) and a low aerosol number before adding organic NPF (i.e., Inorg simulation) in these regions. The enhancement in nucleation rates due to the



inclusion of organic nucleation is more significant in the PD experiment (39%) compared to the PI experiment (6%) (Fig. 4). This is mainly caused by higher sulfuric acid concentrations in PD environment (Fig. S2), resulting in higher heteromolecular nucleation rates involving sulfuric acid and organics (Figs. S6 and S7). A detailed discussion of the specific reason is provided in Section 4.2. Therefore, comparing to the increase in the ~1.7 nm nucleation rate, the increase in the sub-20 nm growth rate plays a more significant role in greater increase of CCN burden in PI experiment (Fig. 4 and Fig. S13).

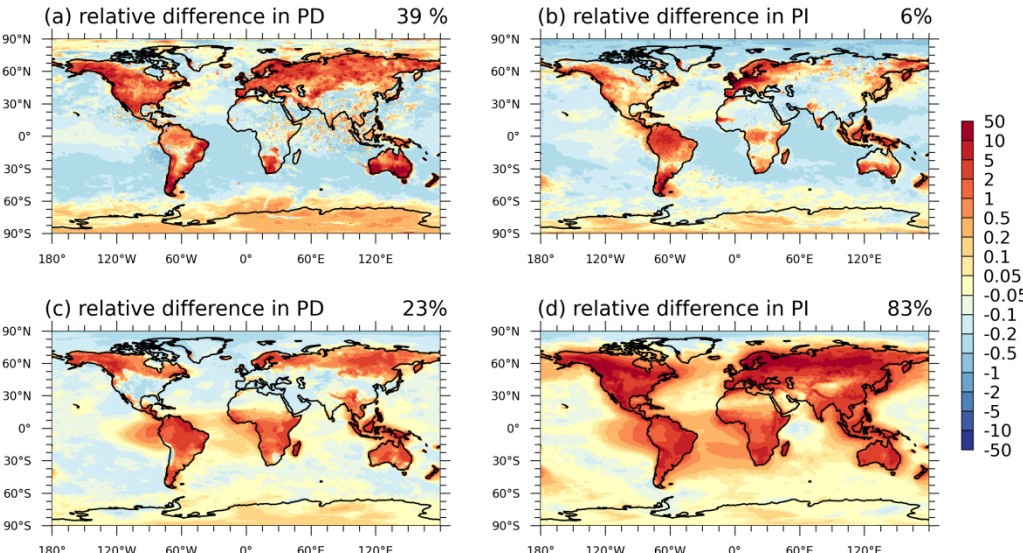

**Figure 4.** Spatial distribution in **(a)** PD_Inorg_Org and **(b)** PI_Inorg_Org (unit: $m^{-2}$ $s^{-1}$). The relative change of the simulated vertically-integrated nucleation rate ($j_{1.7nm}$, below 15 km) and vertically-mean sub-20nm growth rate after adding organic nucleation is shown in PD and PI environments. Global mean values are shown on the top right of each figure.

## 4.2 Effect on aerosol indirect radiative forcing

The significant increase in CCN number and CDNC in the PI experiment resulting from the inclusion of the organic NPF scheme is likely to reduce the aerosol radiative forcing. The aerosol direct radiative forcing may not be significantly influenced by the NPF mechanism because it is not strongly affected by the aerosol size distribution (Rap et al., 2013). Thus, in this study we only quantify the effect of including biogenic organic NPF on the indirect aerosol forcing component ($ERF_{aci}$). The effect of organic NPF on the magnitude of $ERF_{aci}$ is calculated by comparing simulations with (Inorg_Org) and without (Inorg) organic nucleation and growth mechanisms. To analyze the change in $ERF_{aci}$, we also compare the fractional changes of other key variables (nucleation rate, growth rate, aerosol number, CCN number, and CDNC) from PI to PD in Inorg_Org and Inorg (Fig. 7).



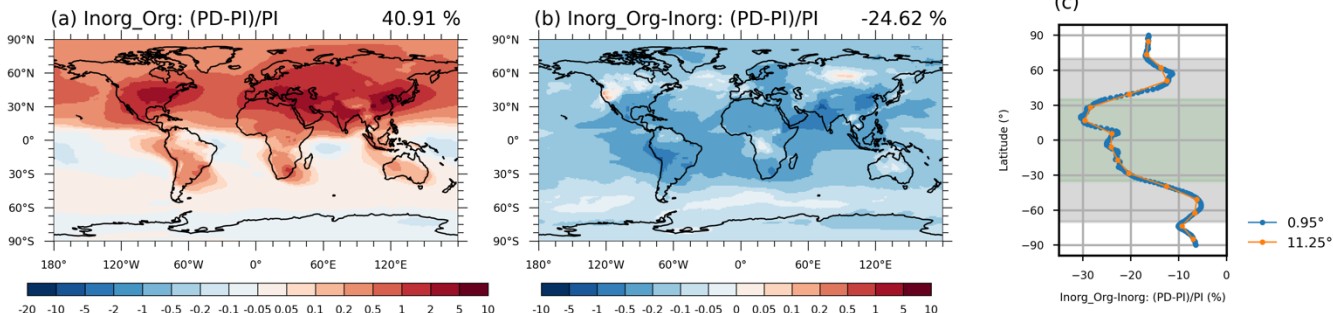

**Figure 5.** The global mean of the relative change of vertically-integrated CCN number at 0.2% supersaturation in PD experiments compared to PI experiments after adding organic NPF mechanisms **(a)** and the difference of that value between with and without organic NPF **(b)**. The zonal mean of the information in **(b)** is shown in **(c)** (Blue scatters show the zonal mean with an interval of 0.95° and the orange scatters show the zonal mean with an interval of 11.25°). The global average value is shown on the top right of each panel. Model experiments are described in **Table 2** and model data come from monthly mean value over 10 years.

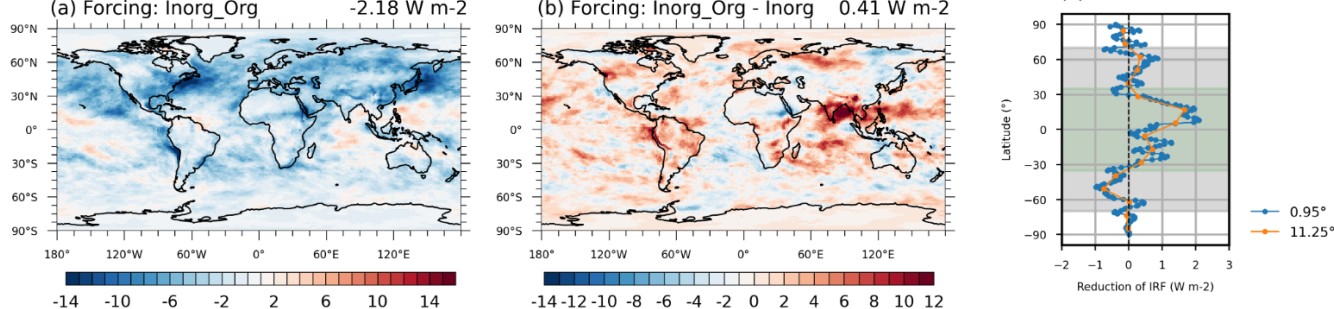

**Figure 6.** The effective radiative forcing due to aerosol–cloud interactions ($ERF_{aci}$) aerosol after including organic NPF **(a)** and the difference in the $ERF_{aci}$ of anthropogenic aerosol between with and without organic NPF **(b)**. The zonal mean of the information in **(b)** is shown in **(c)** (Blue scatters show the zonal mean with an interval of 0.95° and the orange scatters show the zonal mean with an interval of 11.25°). The global average value is shown on the top right of each panel. Model experiments are described in **Table 2** and model data come from monthly mean value over 10 years.

We estimate that the global $ERF_{aci}$ since 1850, after including organic NPF, is -2.18 W m$^{-2}$ (Fig. 6a). The calculated aerosol $ERF_{aci}$ decreases by approximately 0.4 W m$^{-2}$ (corresponding to a 16% reduction) after adding organic NPF mechanisms (Fig. 6b). This reduction is attributed to the greater increase in CCN number in the PI experiment compared to the PD experiment (Fig. 2) when adding organic NPF (Inorg_Org), leading to a smaller relative difference in these variables between the PD and PI experiments (66% in Inorg_Org and 41% in Inorg, Fig. 7).

The largest reduction in $ERF_{aci}$ occurs over the tropical region (-35° to 35° N) (Fig. 6c), especially over oceans with high low cloud cover, such as the western side of the Amazon basin and eastern China (Fig. 6b). This corresponds to the change in the CCN number and CDNC in the PI experiments, which also shows the largest increase in tropical regions. The average $ERF_{aci}$ decreases by approximately 1W m$^{-2}$ between 35°S and 35° N, with a more significant effect in the Northern Hemisphere (NH)



(Fig. 6b and 6c). This is mainly attributed to the largest reduction in the PD fractional change of vertically-integrated CCN number in the northern United States, southeastern United States, and India in Inorg_Org compared to Inorg (Fig. 5b). These significant reductions are transported to the western side of these regions, where most of the anthropogenic aerosol–cloud radiative forcing occurs, resulting in significant reductions in ERF$_{aci}$ (Fig. 6b). The largest reduction of ERF$_{aci}$ in the Southern Hemisphere (SH) occurs on the western side of the Amazon and Australia (Fig. 6b), where biogenic NPF causes the largest reduction in CCN concentrations from PD to PI experiment (Fig. 5a), driven by the large continental source of biogenic gases (Fig. S3). In some tropical oceanic regions of the SH, there are higher CCN concentrations in the PI atmosphere than PD in Inorg_Org (Fig. 5a), caused by higher preindustrial ACCs (Fig. S3) and lower particle condensation sinks, leading to a positive ERF$_{aci}$ (Fig. 5a). At mid-latitudes (35°∼70°) in the NH, the reduction in ERF$_{aci}$ is mainly caused by larger emissions of monoterpenes, and consequently, higher concentrations of HOMs in the PI environment of boreal forests in North America and Eurasia (Fig. S3). The large increase in sub-20 nm particle growth rate in the PI experiment resulting from HOM condensation also supports this point.

In previous studies (Zhu et al., 2019; Gordon et al., 2016), organic nucleation ($J_{Org,n}+J_{Org,i}+J_{SA-Org}$) is the main reason for higher CCN number in PI simulation and thus leading to the reduction in ERF$_{aci}$. However, in our simulation, the fractional change in total nucleation rate from PI to PD is larger after adding organic nucleation (1075% in Inorg_Org and 796% in Inorg) (Fig. 7). This is mainly caused by heteromolecular nucleation rate of sulfuric acid and organics rate ($J_{SA-Org}$), which is the dominant contributor to total nucleation rate, showing greater increase in the PD experiment (Fig. S6) compared to PI experiment (Fig. S7). Especially in boreal forests, northern America, and Australia (Fig. S8), both H$_2$SO$_4$ and HOMs are abundant in the PD experiments (Fig. S2 and S3), leading to much larger $J_{SA-Org}$ values (Fig. S6). Therefore, the greater enhancement of CCN burden in PI experiment and reduction in ERF$_{aci}$ are mainly caused by organic condensational growth on sub-20nm particles instead of organic nucleation.

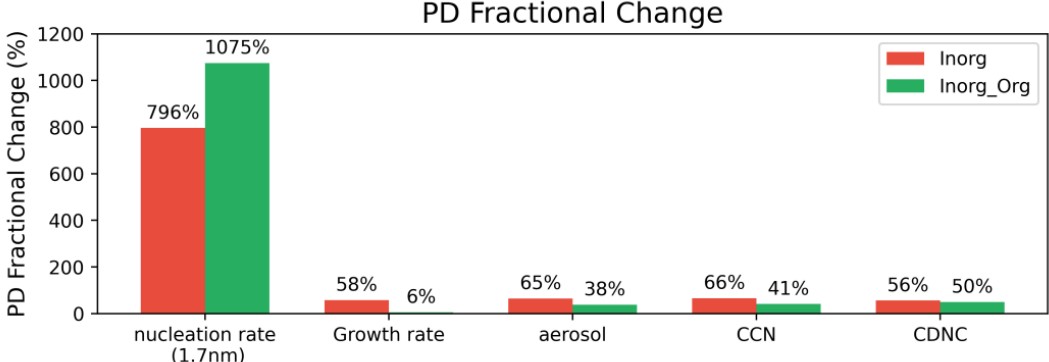

**Figure 7.** The global mean of the PD fractional change ((PD-PI)/PI) of key variables . Model experiments are described in Table 2 and model data come from monthly mean value over 10 years.



The most significant changes in ERF$_{aci}$ reduction due to the inclusion of organic NPF are in tropical regions (Fig. 6c). This is different from previous studies (Zhu et al., 2019; Gordon et al., 2016), which showed that most of the reduction in ERF$_{aci}$ occurs in the mid-latitudes of the NH, closely related to the distribution of HOM concentrations. Previous studies (Zhu et al., 2019; Gordon et al., 2016) did not account for organic nucleating species derived from isoprene oxidation, thereby neglecting

ACC generation through self- and cross-reactions of isoprene- and monoterpene-derived radicals was neglected. Gorden et al. (2016) and Zhu et al. (2019) assumed that all organic nucleating species had the same volatility and can equally contribute to the organic nucleation. This simplification may have led to an overestimation of the organic nucleation rate and, consequently, the reduction in ERF$_{aci}$ in the mid-latitudes. Our study highlights that only ACCs can contribute to pure organic nucleation due to their extremely low volatility. ACCs show high concentrations only in the Amazon, Central Africa and Western Europe

(Fig. S3) where the total nucleation rate is dominated by pure organic nucleation (Figs. S6 and S7). Furthermore, in these regions, there is the largest difference in ACCs concentration between PD and PI experiments (Fig. S3). Consequently, the most significant reductions in ERF$_{aci}$ are in the Amazon, central Africa, Australia, and Southeast Asia (Fig. 6b), as well as the marine low-cloud regions to the west of these areas.

## 5 Summary and discussion

New particle formation (NPF) is widely recognized as an important source of atmospheric particles that significantly influence the Earth's climate. In the present work, the contribution of highly oxygenated organic molecules (HOMs) to cloud condensation nuclei (CCN) burden via organic nucleation is quantified in both present-day (PD) and preindustrial (PI) environments using a chemistry-climate model (Shao et al., 2024). The reduction in effective radiative forcing due to aerosol–cloud interactions (ERF$_{aci}$) caused by adding organic NPF mechanisms is also assessed.

After incorporating organic NPF scheme with state-of-the-art chemical mechanisms for biogenic HOMs into CAM6-Chem, the simulated CCN numbers agree better with measurements across different backgrounds (Fig. 1). Globally, the inclusion of organic-related NPF processes results in a 39% increase in CCN burden in the PI experiment and an 18% increase in the PD experiment. Similarly, cloud droplet number concentration (CDNC) at the top of low clouds in the Inorg_Org simulation rises

by 12% in the PI experiment but only by 7% in the PD experiment. The greater enhancement of CCN burden in the PI experiment is primarily driven by organic condensational growth on sub-20 nm particles, rather than organic nucleation. The main reason is that sulfuric acid (H$_2$SO$_4$) concentrations are significantly higher in the PD environment (Fig. S2), leading to higher heteromolecular nucleation rates of sulfuric acid and organics ($J_{SA-Org}$), which is the largest contributors to organic nucleation in most regions. In contrast, HOM concentrations are higher in the PI atmosphere (Fig. S3) leading to a much

greater condensation of organics on sub-20 nm particles in PI experiments



The larger increase in both CCN and CDNC in the PI environment directly leads to an increased aerosol indirect effect in the PI, which is the baseline for calculating $ERF_{aci}$ in the global model, thereby decreasing the cooling effect of $ERF_{aci}$ (~ 0.42 W m$^{-2}$, corresponding to 16% of its original magnitude) (Fig. 9). The reduction in the magnitude of $ERF_{aci}$ is primarily concentrated in boreal forests and low latitudes (Amazon, central Africa, and Southeast China), consistent with the higher increase in CCN and cloud droplet numbers in the PI atmosphere of those regions.

Although we utilized explicit chemical reactions to replace the traditional fixed yield method for simulating biogenic HOMs concentrations, further studies are needed to better align simulated HOM concentrations with widespread measurements. For instance, the autoxidation reaction step, which likely affects the volatility of the final products and their contribution to organic nucleation, has not yet been conclusively determined in chamber experiments (Roldin et al., 2019; Weber et al., 2020; Berndt et al., 2018). Also, the mechanisms used in this study cannot yet capture all variations in observed NPF events (Shao et al., 2024), especially in polluted environments. More lab-based studies are needed to examine the chemical reactions of anthropogenic HOMs to identify which could contribute to NPF mechanisms.

Despite the many uncertainties in NPF mechanisms, we present the estimated weakening of $ERF_{aci}$ caused by incorporating state-of-the-art biogenic NPF mechanisms in the global model, with a greater reduction than estimated in previous studies. This finding implies that the climate is more sensitive to biogenic emissions and less sensitive to anthropogenic emissions than previously thought. This insight could be broadly applicable in the context of future carbon neutrality when anthropogenic gases and aerosols will decrease, but biogenic emissions will likely increase..

**Competing interests.** At least one of the (co-)authors is a member of the editorial board of Atmospheric Chemistry and Physics.

**Acknowledgments.** This work is supported by the National Natural Science Foundation of China (grant nos. 41925023, U2342223, and 91744208), and the Fundamental Research Funds for the Central Universities - CEMAC "GeoX" Interdisciplinary Program (2024ZD05) by the Frontiers Science Center for Critical Earth Material Cycling, Nanjing University. We greatly thank the High Performance Computing Center (HPCC) of Nanjing University for providing the computational resources used in this work. We thank all the scientists, software engineers, and administrators, who contributed to the development of CESM2.

**Author contributions.** MW and XD designed the study. XS performed the data analysis, produced the figures, and wrote the manuscript draft. LR and MY collected the dataset. YL, SA, WS, HW, JW, XZ and KS contributed to the analysis methods. DJ provided the model. All the authors contributed to discussion, writing, and editing of the manuscript.



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
