# Peer review of "The effect of organic nucleation on the indirect radiative forcing with a semi-explicit chemical mechanism for highly oxygenated organic molecules (HOMs)"

_EGUsphere, 2024_

## Referee Comment (RC1)

"The effect of organic nucleation on the indirect radiative forcing with a semi-explicit chemical mechanism for highly oxygenated organic molecules (HOMs)" by Shao et al. is a science development that evaluates the aerosol, cloud, and radiative forcing changes in the CAM-Chem model using a newly developed organic chemistry mechanism from Xu et al. (2022) and a new HOM nucleation scheme developed by Shao et al. (2024). The paper is succinct and has an overall organized structure, although I think some key information on the critical insights should be discussed earlier in the introduction section instead of in Sect. 4 (see the second comment below), especially because this insight appears throughout the whole manuscript but is not explained early on. Many comments of mine are requests for clarification. I would suggest major revision for this paper, which can be accepted after addressing the comments below. The manuscript can also be benefited by some rewriting to improve grammar and readability.

Lines 28-29: The readability of the sentence is not good since the reader is not clear what Gordon 2016 concluded. It may make the sentence clearer by rephrasing the sentence as "The reduction is mainly driven by ..., instead of the findings of Gordon et al. (2016) that the ~1 nm nucleation drives the reduction."

Line 48: new particle formation can be replaced by NPF.

Lines 72-73: It is not clear to the reader why there is a reduction in the magnitude of the radiative forcing. This is explained at the very end of the manuscript in Sect 4, but readers are confused here, so the importance of HOM-driven NPF in line 68 could be illustrated here in line 72. Some of the information in lines 280-285 could be covered here.

Lines 81-83: The sentence is not clear here by themselves. Reader is curious and wants more details on what the specific advances in Xu 2022 and Shao 2024 are, and why they are better than Vehkamaki 2002, Gordon 2016, and other relevant paper. Please use a few sentences here to give a summary. You have more details in Sect. 2, but I think more details are needed here as well.

Lines 177-178: Why is H2SO4 overestimated over Brazil, Barrow, and Graciosa but not other places? Please also explain why H2SO4 concentration is overestimated in CAM-Chem.

Line 179-180: It is not clear whether the Shao 2024 organic nucleation scheme (Inorg_Org) has increased CCN across the whole globe compared to the Inorg run. It looks like some

places have worsened underestimations (such as Azores for 0.5%SS and 1%SS), which requires an explanation.

Fig. 1: According to Fig. 1, the improvements of the Shao 2024 organic nucleation scheme (Inorg_Org) is not very significant and distinguishable from the Inorg run. It does not appear to the reader that the data points are significantly close to the 1:1 line or the NMB is largely reduced. Also, reader is curious about the correlation coefficient, which might not show significant improvements either. Please include correlation coefficients for all panels.

Multiple measurements still have predictions several orders of magnitude off from the 1:1 line. Overall many data points are stacked together so it is hard to examine the comparison region by region. From reader's perspective, this figure is not showing the merits of the new nucleation scheme.

Lines 185-186: Why does 0.2% SS in Fig. 1b has much fewer CCN number concentrations than other panels in Fig. 1?

Lines 194-195: This sentence does not make much sense to reader. Why would a CCN map (Fig. 2) with a much wider area coverage than the coverage of an aerosol burden map (Figs. S11-S12) be due to the slower removal rate of larger aerosol particles? Please rewrite and clarify.

Lines 254-255: Do you have mode-specific values to let readers understand the changes in the indirect forcings for each mode?

Lines 81-83: The sentence is not clear here by themselves. Reader is curious and wants more details on what the specific advances in Xu 2022 and Shao 2024 are, and why they are better than Vehkamaki 2002, Gordon 2016, and other relevant paper. Please use a few sentences here to give a summary. You have more details in Sect. 2, but I think more details are needed here as well.

---

## Referee Comment (RC2)

Review of "The effect of organic nucleation on the indirect radiative forcing with a semi-explicit chemical mechanism for highly oxygenated organic molecules (HOMs)" by Shao et al. [Research Article, egusphere-2024-4135]

This is an interesting study. The authors incorporated a HOMs-related chemistry and nucleation scheme into a climate model and explored the influence of the HOMs-derived NPF on CCN formation, along with the ensued changes in radiative forcings caused by ACI. Their results showed that including NPF mechanisms in the model can improve simulations of CCN number concentrations. With the new scheme, the authors found that significantly more CCN are generated from organic NPF in PI than PD, which greatly weakens the effective radiative forcing due to ACI. They argued that the weakened radiative forcings are not caused by altered nucleation rates as proposed by Gordon et al. (2016). Rather, it is primarily driven by the more enhanced sub-20 nm growth rate in PI compared to PD. I enjoyed reading this paper, which is very well organized and easy to follow. But there are a few grammatical errors that should be corrected. I have listed some below, but not exhaustive. This work demonstrates the important role of biogenic NPF in CCN formation especially in PI, which enhances our understanding of aerosol emissions and associated nucleation processes in radiative forcings and might help reduce the uncertainties of ACI among climate models in the future. Therefore, I believe this paper will be well-suited for publication in ACP if a few minor issues are addressed.

1. An important finding of this study is that the greater increase in CCN in PI than PD is attributed to organic condensational growth on sub-20 nm particles. I wonder whether the authors have examined the full CCN budget to rule out the impact of other contributing processes. For example, the sinks of CCN via precipitation scavenging might also play an important role.

2. To minimize the influence of meteorological fields on the results, the authors nudged T, winds, and others to MERRA2 reanalysis in the short-term simulations. But the methodology of the nudging experiment is not clearly described, such as what nudging time scale and temporal resolution of MERRA2 are used. A brief examination of simulated winds fields against MERRA2 would also be helpful to strength confidence in the nudging approach.

3. Since the primary goal of Figure 1 is to compare the bias errors of two simulation experiments against observations, rather than to examine correlations, I strongly recommend replacing the scatter plots with bar plots. The current scatter plots do not effectively convey the contrast between the two experiments. Instead, bar plots would allow a clearer demonstration of the simulation errors for each experiment across different regions. To enhance interpretability, the regions could be categorized into Marine, Urban, and Mountain environments, using distinct colors for each category. Additionally, to examine whether the improvement in CCN simulations by Inorg_Org is statistically significant, error bars should be added also.

**Specific comments:**
L30-32: The authors may consider removing the sentences "while the greater … nucleation rates involving sulfuric acid and organics". The current sentences look confusing here and disrupt the

reading flow. Alternatively, if the authors wish to retain these sentences, they need to rephrase these sentences to more clearly align with the argument proposed by Gordon et al. (2016).

L57: "are they" to "they are"

L59: "stringent" to "rigorous"

L171-172: It is difficult to discern the regional variation in bias from Figure 1.

L213: "simulated" to "found"

L215: "a low" to "the originally low"

L221: "in PI" to "in the PI"

L224: Add description for panels (c) and (d)

L229: CCN number concentration?

L230: But adding the NPF mechanism would increase aerosol burden, and thus change the direct radiative forcing, although the size distribution might not change.

L278: "in PI" to "in the PI"

L278: "leading" to "leads"

L281: "compared to PI" to "compared to the PI"

L284: "in PI" to "in the PI"

L295: remove "was neglected"

L312: Be specific about backgrounds

L315-320: To be clearer, the authors should first clarify how organic nucleation changes are responsible for greater enhancement in PI's CCN burden in previous studies or Gordon et al. (2016). They can then highlight how their findings differ from those earlier results.

L320: add a period at the end of the sentence.

L328-330: Rephase the sentences to emphasize "although our methods improve the simulations of CCN burdens"

L336: remove "the"

L340: remove the extra period

---

## Author Comment (AC1)

We are very grateful to the evaluations from the reviewers, which have allowed us to clarify and improve the manuscript. Below we addressed the reviewer comments, with the reviewer comments in black and our response in **blue**.

**Reply for the referee comment#1**
**General Comments:**

"The effect of organic nucleation on the indirect radiative forcing with a semi-explicit chemical mechanism for highly oxygenated organic molecules (HOMs)" by Shao et al. is a science development that evaluates the aerosol, cloud, and radiative forcing changes in the CAM-Chem model using a newly developed organic chemistry mechanism from Xu et al. (2022) and a new HOM nucleation scheme developed by Shao et al. (2024). The paper is succinct and has an overall organized structure, although I think some key information on the critical insights should be discussed earlier in the introduction section instead of in Sect. 4 (see the second comment below), especially because this insight appears throughout the whole manuscript but is not explained early on. Many comments of mine are requests for clarification. I would suggest major revision for this paper, which can be accepted after addressing the comments below. The manuscript can also be benefited by some rewriting to improve grammar and readability.

**Response**: We would like to thank the referee for providing the insightful suggestions, which indeed help us further improve the manuscript. We have added required discussion to account for the major and minor comments and marked the corresponding line number in the revised paper. Please see the revision and the response for the comments as follows.

**Major Comment#1:** Lines 28-29: The readability of the sentence is not good since the reader is not clear what Gordon 2016 concluded. It may make the sentence clearer by rephrasing the sentence as "The reduction is mainly driven by …, instead of the findings of Gordon et al. (2016) that the ~1 nm nucleation drives the reduction."

**Response**: We apologize for any confusion caused by the original sentence. The sentence in Lines 28-29 were modified as (The underlined content is newly added or modified):

"~~Unlike the findings of Gordon et al. (2016), the reduction is mainly driven by a greater enhancement of the sub-20 nm growth rate (GR) in the PI atmosphere compared to PD instead of the ~1 nm nucleation rate ($j_{1.7nm}$) .~~ The reduction is mainly driven by a greater enhancement of the sub-20 nm growth rate (GR) in the PI atmosphere compared to PD, instead of the findings of Gordon et al. (2016) that the ~1 nm nucleation rate ($j_{1.7nm}$) drives the reduction."

**Major Comment#2:** Line 48: new particle formation can be replaced by NPF.
**Response**: Done. Thanks.

**Major Comment#3:** Lines 72-73: It is not clear to the reader why there is a reduction in the magnitude of the radiative forcing. This is explained at the very end of the manuscript in Sect 4, but readers are confused here, so the importance of HOM-driven NPF in line 68 could be illustrated here in line 72. Some of the information in lines 280-285 could be covered here.

**Response**: To improve clarity, we have revised lines 72-73 to better explain the reduction in radiative forcing. This revision incorporates the importance of HOM-driven NPF and integrates key points from lines 280-285.

The sentence in Lines 68-73 were modified as (The underlined content is newly added or modified):

"HOM-driven NPF is especially important in the pristine PI atmosphere, where concentrations of sulfuric acid and ammonia were much lower. Simulations of the pre-industrial atmosphere form the baseline for calculating anthropogenic radiative forcing in global models (Carslaw et al., 2013), where higher monoterpene emissions led to greater HOM concentrations, thereby enhancing nucleation and particle growth. Using global model simulations, Gordon et al. (2016) showed that new particles formed from monoterpene-derived HOMs could increase CCN concentrations in the PI environment by 20% to 100%, a rise considerably larger than the increase simulated for PD conditions. This leads to a 27% reduction in negative radiative forcing  since 1750, decreasing by between -0.28 W m$^{-2}$ and -0.06 W m$^{-2}$."

**Major Comment#4:** Lines 81-83: The sentence is not clear here by themselves. Reader is curious and wants more details on what the specific advances in Xu 2022 and Shao 2024 are, and why they are better than Vehkamaki 2002, Gordon 2016, and other relevant paper. Please use a few sentences here to give a summary. You have more details in Sect. 2, but I think more details are needed here as well.

**Response**: Sorry for confusion caused by the original sentence. We have revised Lines 81-83 to provide more detailed information on the advances in Xu et al. (2022) and Shao et al. (2024). The sentence in Lines 81-83 were modified as (The underlined content is newly added or modified):

"Considering the unequivocal evidence for the role of biogenic organics in producing atmospheric particles, Shao et al. (2024) have recently incorporated a state-of-the-art representation of HOMs from various chamber experiments (Xu et al., 2022). This representation semi-explicitly treats the unimolecular autoxidation of monoterpene-derived $RO_2$ radicals and their self- and cross-reactions with other $RO_2$ species, rather than using the empirical fixed HOM yield (Gordon et al., 2016; Zhao et al., 2018). In addition, Shao et al. (2024) introduced a HOM-involving nucleation parameterization (Riccobono et al., 2014; Kirkby et al., 2006) and enabled these organics to condense onto newly formed sub-20 nm particles. ."

**Major Comment#5:** Lines 177-178: Why is $H_2SO_4$ overestimated over Brazil, Barrow, and Graciosa but not other places? Please also explain why $H_2SO_4$ concentration is overestimated in CAMChem.

**Response**: Although direct observations of $H_2SO_4$ are not globally available, existing evaluations indicate that CAM-Chem systematically overestimates its concentration across many regions (He et al., 2014, Shao et al., 2024), not only in Brazil, Barrow, and Graciosa. This is a feature of CAM6, as evidenced by comparisons with previous model simulations (Table S6 in Shao et al. 2024) and measurements (Table S7 in Shao et al. 2024), which will be presented below.

We highlight that this overestimation has the largest impact in Brazil, Barrow, and Graciosa. At these urban sites the growth of newly formed particles is affected mainly by the condensation of $H_2SO_4$ onto sub-20 nm particles (see Shao et al., 2024, Fig. 7), so the overestimation of $H_2SO_4$ concentration directly amplifies the simulated growth rate. In rural and mountainous regions abundant monoterpenes supply organic vapours that dominate particle growth, so the $H_2SO_4$ bias has only a minor effect there.

**Table S5 (from Shao et al., 2024).** Comparison of Gas Burden (Tg S) across different studies

| | This study (CESM-MAM4) | Liu et al. (2012) | | Spracklen et al. (2005) | Mann et al. (2010) |
|---|---|---|---|---|---|
| | Inorg_Org | CESM-MAM3 | CESM-MAM7 | GLOMAP-bin | GLOMAP-mode |
| $SO_2$ | 0.29 | 0.35 | 0.34 | 0.49 | 0.3 (0.2-0.68) |
| DMS | 0.067 | 0.067 | 0.067 | 0.04 | 0.027 (0.02-0.15) |
| $H_2SO_4$ | 0.00054 | 0.0004 | 0.00042 | / | 0.0001 |

**Table S6 (from Shao et al., 2024).** Comparison of $H_2SO_4$ concentration with measurements

| | $[H_2SO_4]$ $10^6$ mole cm$^{-3}$ | | | |
|---|---|---|---|---|
| Measurement site | Mean | | Median | |
| | Measurement | Simulation | Measurement | Simulation |
| Hyytiälä, Finland | 0.43 | 2.61 | 0.18 | 0.3 |
| San Pietro Capofiume, Italy | 5.40 | 4.57 | 2.40 | 2.79 |
| Melpitz, Germany | 6.43 | 10.53 | 2.94 | 2.8 |
| Niwot Ridge, Colorado USA | 1.83 | 2.52 | 1.40 | 1.36 |
| Atlanta, Georgia USA | 12.90 | 9.95 | 2.85 | 1.03 |
| Beijing, China | 2.51 | 17.91 | 1.81 | 3.15 |

In CAM-Chem the overestimation of gas-phase $H_2SO_4$ originates from several reasons. Anthropogenic emissions of the $SO_2$ are likely too high, wet scavenging removes $SO_2$ too inefficiently, and the representation of in-cloud aqueous chemistry is overly simplified. The overestimation of $SO_2$ emission in CAM-Chem produced a normalized mean bias of 219.1 % for $SO_2$ concentration, rising to 244.7 % at CASTNET sites (He et al., 2015). Sensitivity experiments by He and Zhang (2014) show that reducing $SO_2$ emissions by 30 % lowers the simulated bias from 291.8 % to 152.2 %. Ge et al. (2021, 2022) further demonstrated that incorporating detailed in-cloud aqueous-phase chemistry and enhanced sulfate wet deposition significantly improves the simulation of $SO_2$ and $H_2SO_4$. In addition, He et al. (2015) reported that CAM-Chem tends to

underestimate precipitation in the eastern United States, which weakens wet scavenging of $SO_2$ and amplifies the overestimation of $H_2SO_4$.

The sentence in Lines 177-178 were modified as (The underlined content is newly added or modified):

"Apart from urban regions like Manacapuru, Brazil, the overestimation also increases over oceanic regions such as Barrow and Graciosa. These overestimations in CCN numbers in the Inorg model are likely related to the overestimation of $H_2SO_4$ concentration in CAM6-Chem (Shao et al., 2024), as these regions are more sensitive to $H_2SO_4$ due to the limited presence of organic NPF precursors like monoterpenes. The overestimation of $H_2SO_4$ concentrations in the CAM-Chem model is likely the result of multiple contributing factors, such as overestimated $SO_2$ emissions (He et al., 2014; He and Zhang, 2014), insufficient representation of in-cloud chemistry (Ge et al., 2021; 2022), underestimated wet deposition processes (He et al., 2015; He and Zhang, 2014)."

**Major Comment#6:** Line 179-180: It is not clear whether the Shao 2024 organic nucleation scheme (Inorg_Org) has increased CCN across the whole globe compared to the Inorg run. It looks like some places have worsened underestimations (such as Azores for 0.5%SS and 1%SS), which requires an explanation.

**Response**: The organic NPF scheme in Shao et al. (2024) (Inorg_Org) does increase the global CCN burden compared to the Inorg simulation, but not necessarily the surface number concentration. The CCN values shown for the Azores in Figure 1 represent surface concentrations (Table 1); therefore, it is possible that the underestimation becomes more pronounced in the Inorg_Org simulation after including the organic NPF scheme.

The main reason for the worsened underestimations in Azores (ocean region) at 0.5% SS and 1% SS may be that, after including organic NPF, heteromolecular nucleation involving $H_2SO_4$ and organics ($J_{SA-Org}$) consumes more $H_2SO_4$ over land, as monoterpene emissions and the formation of HOMs are mainly concentrated over there (Fig. S3). As a result, less $H_2SO_4$ is transported to the ocean, where nucleation (Figs. S4 and S5) and sub-20 nm particle growth are primarily driven by $H_2SO_4$ (Fig. 7 in Shao et al., 2024), leading to decreased nucleation and growth rates over ocean.

In light of this, the sentence in Line 219 were modified as (The underlined content is newly added):

"The enhancement in nucleation rates due to the inclusion of organic nucleation is more significant in the PD experiment (39%) compared to the PI experiment (6%) (Fig. 4). This is mainly caused by higher sulfuric acid concentrations in the PD environment (Fig. S2), resulting in higher heteromolecular nucleation rates involving sulfuric acid and organics ($J_{SA-Org}$) (Figs. S6 and S7) over

land, where both $H_2SO_4$ and HOMs show high values. Consequently, more $H_2SO_4$ is consumed over land (Fig. S16), reducing its transport to oceanic regions (Fig. S17). As a result, nucleation rates decrease over the ocean in both PD and PI experiments (Fig. 4)."

[Figure]

**Figure S16.** Spatial distribution of the simulated vertically-integrated sulfuric acid nucleation loss rate in (a) PD and (b) PI experiments (unit: ng m$^{-2}$ s$^{-1}$). The relative difference between Inorg_Org and Inorg in PD and PI experiments is shown in (c) and (d) (unitless). Global mean values are shown on the top right of each figure.

[Figure]

**Figure S17.** Spatial distribution of the simulated vertically-integrated sulfuric acid concentration in (a) PD and (b) PI experiments (unit: cm$^{-2}$). The difference between Inorg_Org and Inorg in PD and PI experiments is shown in (c) and (d) (unitless). Global mean values are shown on the top right of each figure.

**Major Comment#7:** Fig. 1: According to Fig. 1, the improvements of the Shao 2024 organic nucleation scheme (Inorg_Org) is not very significant and distinguishable from the Inorg run. It does not appear to the reader that the data points are significantly close to the 1:1 line or the NMB is largely reduced. Also, reader is curious about the correlation coefficient, which might not show significant improvements either. Please include correlation coefficients for all panels.

Multiple measurements still have predictions several orders of magnitude off from the 1:1 line. Overall many data points are stacked together so it is hard to examine the comparison region by region. From reader's perspective, this figure is not showing the merits of the new nucleation scheme.

**Response**: Thank you for your constructive feedback. We have revised Figure 1 to use bar plots instead of scatter plots to better compare the performance between the two simulations across different regions (categorized into Marine, Urban, Rural, and Mountain environments), suggested by reviewer#2. The original scatter plot has been moved to the Supplementary Information as Figure S18, with correlation coefficients (R) provided for reference.

We have revised the original description in **Section 3**:

**Original Paragraph:**
"CCN number concentrations in the Inorg_Org model at 0.1%, 0.2%, 0.5%, and 1% supersaturation (ss) show better agreement with measurements from various locations compared to the Inorg model (Fig. 1). The underestimation of CCN numbers in the Inorg simulation is alleviated by incorporating organic-related NPF, especially over rural and mountainous regions such as Steamboat Springs, Shouxian, and Nainital (Fig. 1), where both nucleation and initial growth rates are dominated by biogenic pathways. However, there remains a slight underestimation of CCN number in Shouxian at all supersaturation levels. This is likely due to the neglect of anthropogenic-derived HOMs to nucleation and growth, which are key NPF mechanisms in rural regions of China. The increase in CCN number due to the addition of organic NPF mechanisms is simulated not only in the locations listed in Table 1 but also on a global scale (see Fig. 7). "In urban regions of Brazil (Manacapuru), the overestimation of CCN numbers at 0.5% and 1% ss in Inorg is exacerbated (Fig. 1). Apart from urban regions like Manacapuru, Brazil, the overestimation also increases over oceanic regions such as Barrow and Graciosa. These overestimations in CCN numbers in the Inorg model are likely related to the overestimation of $H_2SO_4$ concentration in CAM6-Chem (Shao et al., 2024). Overall, the normalized mean bias (NMB) of CCN numbers at different supersaturation levels decreases from -35% (Inorg) to -24% (Inorg_Org), indicating that the Inorg_Org model provides a more accurate representation of organic contributions for further quantification in Section 3."

**Revised Paragraph:**

"The underestimation of CCN numbers in the Inorg simulation is alleviated by incorporating organic-related NPF, especially over rural and mountainous regions (Fig. 1), where both nucleation and initial growth rates are dominated by biogenic pathways. The remaining underestimation of CCN in rural regions (Fig. 1) is likely due to the neglect of anthropogenic-derived HOMs, which may play a key role in NPF in these areas. The increase in CCN number due to the addition of organic NPF mechanisms is simulated not only in the locations listed in Table 1 but also on a global scale (see Fig. 7). In urban regions, the overestimation of CCN numbers is exacerbated (Fig. 1). These overestimations in CCN numbers in the Inorg model are likely related to the overestimation of $H_2SO_4$ concentration in CAM6-Chem (Shao et al., 2024). Overall, the relative bias of CCN numbers at different supersaturation levels decreases from -57% (Inorg) to -45% (Inorg_Org) (Fig. S18), indicating that the Inorg_Org model provides a more accurate representation of organic contributions for further quantification in Section 3."

[Figure]

**Figure 1.** Box plots showing the relative bias (%) between simulated monthly mean and observed median CCN number concentrations across categorized background sites (Marine, Urban, Mountain, Rural). Red and green boxes represent the Inorg and Inorg_Org experiments, respectively. Black triangles indicate the mean relative bias for each category. Numerical values above the boxes denote the corresponding mean normalized mean bias (NMB) for each experiment. Information on the measurement sites is provided in Table 2.

[Figure]

**Figure S18.** Comparison of simulated monthly mean and observed median CCN number concentrations (unit: cm$^{-3}$) at categorized background sites. Results from the Inorg_Org experiment are shown in green, and those from the Inorg experiment are shown in red. Information on the measurement sites is provided in Table 2. Normalized mean bias (NMB) and correlation values are indicated in the top-left corner of each panel.

**Major Comment#8:** Lines 185-186: Why does 0.2% SS in Fig. 1b has much fewer CCN number concentrations than other panels in Fig. 1?

**Response**: We apologize for the confusion. Please note that the panels in Fig. 1 are not plotted on the same scale, which may give the impression that CCN number concentrations at 0.2% SS are much lower. In general, a lower supersaturation results in fewer particles being activated as CCN, as predicted by Köhler theory, compared to higher supersaturation levels. The revised figure and corresponding description have already been provided in response to the **Major Comment#7.**

**Major Comment#9:** Lines 194-195: This sentence does not make much sense to reader. Why would a CCN map (Fig. 2) with a much wider area coverage than the coverage of an aerosol burden map (Figs. S11-S12) be due to the slower removal rate of larger aerosol particles? Please rewrite and clarify.

**Response**: The original text of lines 194-195 is:

"The spatial pattern of change in CCN burden (Fig. 2) is consistent with the change in aerosol burden in the Aitken and accumulation mode (Figs. S11 and S12) but affects much wider areas because of the slower removal rate of larger aerosol particles."

This means the lifetime of CCN (cloud condensation nuclei) is typically longer than that of aerosol particles and is less easily removed (Pierce et al., 2009; Riemer et al., 2009). As a result, CCN can be transported over longer distances and influence a wider area, which explains why the spatial extent of CCN impact is greater than that of aerosol burden.

To avoid further misunderstanding, sentences in Lines 194-195 were modified as (The underlined content is newly added or modified):

"The spatial pattern of change in CCN burden (Fig. 2) is consistent with the change in aerosol burden in the Aitken and accumulation mode (Figs. S11 and S12) but affects much wider areas. because of the slower removal rate of larger aerosol particles. Since ultrafine particles (< 50 nm) are quickly lost by coagulation to larger, pre-existing aerosol, CCN typically have a longer atmospheric lifetime and are less efficiently removed than smaller aerosol particles, allowing them to exert influence over wider spatial scales (Pierce et al., 2009; Riemer et al., 2009)."

**Major Comment#10:** Lines 254-255: Do you have mode-specific values to let readers understand the changes in the indirect forcings for each mode?

**Response**: Thank you for your valuable comment. Although the model output does not provide mode-specific values for changes in indirect radiative forcing, we have provided supplementary data (Figure S19) on the particle number size distribution (PNSD) by mode. We hope this will help readers better understand the contributions of different aerosol modes to the overall indirect forcing changes.

Although the number concentration of Aitken mode aerosols exhibits substantial changes after incorporating organic NPF (Fig. S19), only particles larger than 50 nm in diameter can typically activate as CCN under ambient supersaturations (Dusek et al., 2006). Therefore, the change in indirect radiative forcing is mainly dominated by the accumulation mode, as it is the only mode showing significant number concentration changes within the CCN-relevant size range (Fig. S19).

The sentence in Line 255 was modified as (The underlined content is newly added or modified):

"The calculated aerosol $ERF_{aci}$ decreases by approximately 0.4 W m$^{-2}$ (corresponding to a 16% reduction) after adding organic NPF mechanisms (Fig. 6b), primarily driven by changes in the number concentration of accumulation mode aerosols (Fig. S19)."

[Figure]

**Figure S19.** Particle number size distribution (PNSD) by mode in the (a) PD and (b) PI experiments. Solid lines represent the Inorg_Org simulation, and dashed lines represent the Inorg simulation. Each color corresponds to a specific mode: Accumulation (red), Aitken (green), Coarse (orange), and Primary (brown). The shaded blue area highlights particles with diameters larger than 50 nm, which are typically large enough to activate as CCN under ambient supersaturation conditions (Dusek et al., 2006).

**Reference**

Carslaw, K. S., Lee, L. A., Reddington, C. L., Pringle, K. J., Rap, A., Forster, P. M., Mann, G. W., Spracklen, D. V., Woodhouse, M. T., Regayre, L. A., and Pierce, J. R.: Large contribution of natural aerosols to uncertainty in indirect forcing, Nature, 503, 67-71, 10.1038/nature12674, 2013.

Dusek, U. et al. (2006). Size matters more than chemistry for cloud-nucleating ability of aerosol particles. Science, 312(5778), 1375–1378. https://doi.org/10.1126/science.1125261

Dunne, E. M., Gordon, H., Kurten, A., Almeida, J., Duplissy, J., Williamson, C., Ortega, I. K., Pringle, K. J., Adamov, A., Baltensperger, U., Barmet, P., Benduhn, F., Bianchi, F., Breitenlechner, M., Clarke, A., Curtius, J., Dommen, J., Donahue, N. M., Ehrhart, S., Flagan, R. C., Franchin, A., Guida, R., Hakala, J., Hansel, A., Heinritzi, M., Jokinen, T., Kangasluoma, J., Kirkby, J., Kulmala, M., Kupc, A., Lawler, M. J., Lehtipalo, K., Makhmutov, V., Mann, G., Mathot, S., Merikanto, J., Miettinen, P., Nenes, A., Onnela, A., Rap, A., Reddington, C. L. S., Riccobono, F., Richards, N. A. D., Rissanen, M. P., Rondo, L., Sarnela, N., Schobesberger, S., Sengupta, K., Simon, M., Sipilaa, M., Smith, J. N., Stozkhov, Y., Tome, A., Trostl, J., Wagner, P. E., Wimmer, D., Winkler, P. M., Worsnop, D. R., and Carslaw, K. S.: Global atmospheric particle formation from CERN CLOUD measurements, Science, 354, 1119-1124, 10.1126/science.aaf2649, 2016.

Ge, W., Liu, J., Xiang, S., Zhou, Y., Zhou, J., Hu, X., Ma, J., Wang, X., Wan, Y., Hu, J., Zhang, Z., Wang, X., and Tao, S.: Improvement and Uncertainties of Global Simulation of Sulfate Concentration and Radiative Forcing in CESM2, Journal of Geophysical Research: Atmospheres, 127, e2022JD037623, https://doi.org/10.1029/2022JD037623, 2022.

Ge, W., Liu, J., Yi, K., Xu, J., Zhang, Y., Hu, X., Ma, J., Wang, X., Wan, Y., Hu, J., Zhang, Z., Wang, X., and Tao, S.: Influence of atmospheric in-cloud aqueous-phase chemistry on the global simulation of SO2 in CESM2, Atmos. Chem. Phys., 21, 16093-16120, 10.5194/acp-21-16093-2021, 2021.

Gordon, H., Sengupta, K., Rap, A., Duplissy, J., Frege, C., Williamson, C., Heinritzi, M., Simon, M., Yan, C., Almeida, J., Trostl, J., Nieminen, T., Ortega, I. K., Wagner, R., Dunne, E. M., Adamov, A., Amorim, A., Bernhammer, A. K., Bianchi, F., Breitenlechner, M., Brilke, S., Chen, X. M., Craven, J. S., Dias, A., Ehrhart, S., Fischer, L., Flagan, R. C., Franchin, A., Fuchs, C., Guida, R., Hakala, J., Hoyle, C. R., Jokinen, T., Junninen, H., Kangasluoma, J., Kim, J., Kirkby, J., Krapf, M., Kurten, A., Laaksonen, A., Lehtipalo, K., Makhmutov, V., Mathot, S., Molteni, U., Monks, S. A., Onnela, A., Perakyla, O., Piel, F., Petaja, T., Praplanh, A. P., Pringle, K. J., Richards, N. A. D., Rissanen, M. P., Rondo, L., Sarnela, N., Schobesberger, S., Scott, C. E., Seinfeldo, J. H., Sharma, S., Sipila, M., Steiner, G., Stozhkov, Y., Stratmann, F., Tome, A., Virtanen, A., Vogel, A. L., Wagner, A. C., Wagner, P. E., Weingartner, E., Wimmer, D., Winkler, P. M., Ye, P. L., Zhang, X., Hansel, A., Dommen, J., Donahue, N. M., Worsnop, D. R., Baltensperger, U., Kulmala, M., Curtius, J., and Carslaw, K. S.: Reduced anthropogenic aerosol radiative forcing caused by biogenic new particle formation, P. Natl. Acad. Sci. USA, 113, 12053-12058, 10.1073/pnas.1602360113, 2016.

He, J. and Zhang, Y.: Improvement and further development in CESM/CAM5: gas-phase chemistry and inorganic aerosol treatments, Atmospheric Chemistry and Physics, 14, 9171-9200, 10.5194/acp-14-9171-

2014, 2014.

He, J., Zhang, Y., Glotfelty, T., He, R., Bennartz, R., Rausch, J., and Sartelet, K.: Decadal simulation and comprehensive evaluation of CESM/CAM5.1 with advanced chemistry, aerosol microphysics, and aerosol-cloud interactions, Journal of Advances in Modeling Earth Systems, 7, 110-141, https://doi.org/10.1002/2014MS000360, 2015.

Kirkby, J., Duplissy, J., Sengupta, K., Frege, C., Gordon, H., Williamson, C., Heinritzi, M., Simon, M., Yan, C., Almeida, J., Trostl, J., Nieminen, T., Ortega, I. K., Wagner, R., Adamov, A., Amorim, A., Bernhammer, A. K., Bianchi, F., Breitenlechner, M., Brilke, S., Chen, X. M., Craven, J., Dias, A., Ehrhart, S., Flagan, R. C., Franchin, A., Fuchs, C., Guida, R., Hakala, J., Hoyle, C. R., Jokinen, T., Junninen, H., Kangasluoma, J., Kim, J., Krapf, M., Kurten, A., Laaksonen, A., Lehtipalo, K., Makhmutov, V., Mathot, S., Molteni, U., Onnela, A., Perakyla, O., Piel, F., Petaja, T., Praplan, A. P., Pringle, K., Rap, A., Richards, N. A. D., Riipinen, I., Rissanen, M. P., Rondo, L., Sarnela, N., Schobesberger, S., Scott, C. E., Seinfeld, J. H., Sipila, M., Steiner, G., Stozhkov, Y., Stratmann, F., Tome, A., Virtanen, A., Vogel, A. L., Wagner, A. C., Wagner, P. E., Weingartner, E., Wimmer, D., Winkler, P. M., Ye, P. L., Zhang, X., Hansel, A., Dommen, J., Donahue, N. M., Worsnop, D. R., Baltensperger, U., Kulmala, M., Carslaw, K. S., and Curtius, J.: Ion-induced nucleation of pure biogenic particles, Nature, 533, 521-+, 10.1038/nature17953, 2016.

Kulmala, M., Lehtinen, K. E. J., and Laaksonen, A.: Cluster activation theory as an explanation of the linear dependence between formation rate of 3nm particles and sulphuric acid concentration, Atmos. Chem. Phys., 6, 787-793, 10.5194/acp-6-787-2006, 2006

Liu, X., Easter, R. C., Ghan, S. J., Zaveri, R., Rasch, P., Shi, X., Lamarque, J. F., Gettelman, A., Morrison, H., Vitt, F., Conley, A., Park, S., Neale, R., Hannay, C., Ekman, A. M. L., Hess, P., Mahowald, N., Collins, W., Iacono, M. J., Bretherton, C. S., Flanner, M. G., and Mitchell, D.: Toward a minimal representation of aerosols in climate models: description and evaluation in the Community Atmosphere Model CAM5, Geosci. Model Dev., 5, 709-739, 10.5194/gmd-5-709-2012, 2012.

Mann, G. W., Carslaw, K. S., Spracklen, D. V., Ridley, D. A., Manktelow, P. T., Chipperfield, M. P., Pickering, S. J., and Johnson, C. E.: Description and evaluation of GLOMAP-mode: a modal global aerosol microphysics model for the UKCA composition-climate model, Geosci. Model Dev., 3, 519-551, 10.5194/gmd-3-519-2010, 2010. Spracklen, D. V., Pringle, K. J., Carslaw, K. S., Chipperfield, M. P., and Mann, G. W.: A global off-line model of size-resolved aerosol microphysics: I. Model development and prediction of aerosol properties, Atmos. Chem. Phys., 5, 2227-2252, 10.5194/acp-5-2227-2005, 2005.

Merikanto, J., Napari, I., Vehkamaki, H., Anttila, T., and Kulmala, M.: New parameterization of sulfuric acid-ammonia-water ternary nucleation rates at tropospheric conditions, J. Geophys. Res.-Atmos., 112, 10.1029/2006jd007977, 2007

Pierce, J. R., & Adams, P. J. (2009). Uncertainty in global CCN concentrations from uncertain aerosol emissions and nucleation rates. Atmospheric Chemistry and Physics, 9(4), 1339–1356.

Riemer, N., West, M., Zaveri, R. A., & Easter, R. C. (2009). Estimating black carbon aging time-scales with a particle-resolved aerosol model. Atmospheric Chemistry and Physics, 9(4), 1339–1356. arxiv.org

Shao, X., Wang, M., Dong, X., Liu, Y., Shen, W., Arnold, S. R., Regayre, L. A., Andreae, M. O., Pöhlker, M. L., Jo, D. S., Yue, M., and Carslaw, K. S.: Global modeling of aerosol nucleation with a semi-explicit chemical mechanism for highly oxygenated organic molecules (HOMs), Atmos. Chem. Phys., 24, 11365-11389, 10.5194/acp-24-11365-2024, 2024.

Sihto, S. L., Kulmala, M., Kerminen, V. M., Dal Maso, M., Petaja, T., Riipinen, I., Korhonen, H., Arnold, F., Janson, R., Boy, M., Laaksonen, A., and Lehtinen, K. E. J.: Atmospheric sulphuric acid and aerosol formation: implications from atmospheric measurements for nucleation and early growth mechanisms, Atmos. Chem. Phys., 6, 4079-4091, 10.5194/acp-6-4079-2006, 2006.

Vehkamaki, H., Kulmala, M., Napari, I., Lehtinen, K. E. J., Timmreck, C., Noppel, M., and Laaksonen, A.: An improved parameterization for sulfuric acid-water nucleation rates for tropospheric and stratospheric conditions, J. Geophys. Res.-Atmos., 107, 10.1029/2002jd002184, 2002.

Xu, R., Thornton, J. A., Lee, B. H., Zhang, Y., Jaeglé, L., Lopez-Hilfiker, F. D., Rantala, P., and Petäjä, T.: Global simulations of monoterpene-derived peroxy radical fates and the distributions of highly oxygenated organic molecules (HOMs) and accretion products, Atmos. Chem. Phys., 22, 5477-5494, 10.5194/acp-22-5477-2022, 2022.

Zhao, Y., Thornton, J. A., and Pye, H. O. T.: Quantitative constraints on autoxidation and dimer formation from direct probing of monoterpene-derived peroxy radical chemistry, P. Natl. Acad. Sci. USA, 115, 12142-12147, 10.1073/pnas.1812147115, 2018.

---

## Author Comment (AC2)

We are very grateful to the evaluations from the reviewers, which have allowed us to clarify and improve the manuscript. Below we addressed the reviewer comments, with the reviewer comments in black and our response in **blue**.

**Reply for the referee comment#2**
**General Comments:**

This is an interesting study. The authors incorporated a HOMs-related chemistry and nucleation scheme into a climate model and explored the influence of the HOMs-derived NPF on CCN formation, along with the ensued changes in radiative forcings caused by ACI. Their results showed that including NPF mechanisms in the model can improve simulations of CCN number concentrations. With the new scheme, the authors found that significantly more CCN are generated from organic NPF in PI than PD, which greatly weakens the effective radiative forcing due to ACI. They argued that the weakened radiative forcings are not caused by altered nucleation rates as proposed by Gordon et al. (2016). Rather, it is primarily driven by the more enhanced sub-20 nm growth rate in PI compared to PD. I enjoyed reading this paper, which is very well organized and easy to follow. But there are a few grammatical errors that should be corrected. I have listed some below, but not exhaustive. This work demonstrates the important role of biogenic NPF in CCN formation especially in PI, which enhances our understanding of aerosol emissions and associated nucleation processes in radiative forcings and might help reduce the uncertainties of ACI among climate models in the future. Therefore, I believe this paper will be well-suited for publication in ACP if a few minor issues are addressed.

**Response**: We sincerely thank the referee for the positive and thoughtful comments, which indeed help us further improve the manuscript. We have carefully reviewed the manuscript and corrected the grammatical issues you noted, along with other minor edits. Please see the revision and the response for the comments as follows.

**Comment#1:** An important finding of this study is that the greater increase in CCN in PI than PD is attributed to organic condensational growth on sub-20 nm particles. I wonder whether the authors have examined the full CCN budget to rule out the impact of other contributing processes. For example, the sinks of CCN via precipitation scavenging might also play an important role.

**Response**: While the initial model output did not include these variables, we have examined all aerosol types that could be activated into CCN (including $SO_4$, SOA, POM, Sea Salt) and separately calculated their dry and wet deposition fluxes in both the PD and PI experiments for reference (Table S5). Compared to the growth rate, the PD fractional change in the dry and wet deposition fluxes did not show significant changes after adding organic NPF (values of "Inorg_Org - Inorg" in Table S5).

The sentence in Line 221 was modified as (The underlined content is newly added or modified):

"Therefore, compared to the increase in the ~1.7 nm nucleation rate, the increase in the sub-20 nm growth rate plays a more significant role in the greater increase of CCN burden in the PI experiment (Fig. 4 and Fig. S13). Other components of the CCN budget show no substantial changes (Table S5), further reinforcing the dominant role of condensational growth."

**Table S5**. Global mean of dry and wet deposition fluxes of different aerosols (including sulfate, secondary organic aerosol, primary organic matter and sea salt) in PD_Inorg_Org experiments. The PD fractional change ((PD-PI)/PI) of these variables in Inorg_Org and Inorg experiments are also shown. These aerosol sinks serve as proxies for CCN removal processes.

| | Aerosol Type | Value | PD Fractional Change | | |
| --- | --- | --- | --- | --- | --- |
| | | PD_Inorg_Org (ng m$^{-2}$ s$^{-1}$) | Inorg_Org | Inorg | Inorg_Org-Inorg |
| dry deposition flux | $SO_4$ | 0.82 | 25.4% | 25.7% | -0.3% |
| | SOA | 0.08 | 30.6% | 32.2% | -1.6% |
| | POM | 0.04 | 54.9% | 32.2% | -1.6% |
| | Sea Salt | 2.29 | -2.4% | -2.5% | 0.1% |
| wet deposition flux | $SO_4$ | 10.62 | 31.2% | 30.1% | 1.1% |
| | SOA | 0.78 | 49.7% | 48% | 1.7% |
| | POM | 0.78 | 74.5% | 73.9% | 0.7% |
| | Sea Salt | 32.60 | -0.1% | 1.1% | -1.2% |

**Comment#2:** To minimize the influence of meteorological fields on the results, the authors nudged T, winds, and others to MERRA2 reanalysis in the short-term simulations. But the methodology of the nudging experiment is not clearly described, such as what nudging time scale and temporal resolution of MERRA2 are used. A brief examination of simulated winds fields against MERRA2 would also be helpful to strength confidence in the nudging approach.

**Response**: We have included a clear explanation of the nudging methodology and supplemented this with a comparison between the simulated meteorological fields and MERRA2 reanalysis data in the supplementary materials (Fig. S20).

The sentence in Line 143 was modified as (The underlined content is newly added):

"In order to compare simulated CCN with measurements, several short-term simulations were performed, in which meteorological fields (temperature and wind profiles, surface pressure, surface stress, surface heat and moisture fluxes) were nudged toward Modern-Era Retrospective analysis for Research and Applications (MERRA2) reanalysis with a relaxation timescale of 6 hr (Kooperman et al., 2012). Meteorological fields are nudged towards the MERRA2 every 0.5 h, which is the same as the physics timestep of the model (Lamarque et al., 2017). The simulated meteorological fields and their deviations from MERRA2 reanalysis are presented in Figure S20."

[Figure]

**Figure S20.** Spatial distribution of (a) simulated and (b) observed temperature (shaded, unit: K) and wind speed (arrows, unit: m s$^{-1}$). Panel (c) shows the difference in temperature between simulation and observation.

**Comment#3:** Since the primary goal of Figure 1 is to compare the bias errors of two simulation experiments against observations, rather than to examine correlations, I strongly recommend replacing the scatter plots with bar plots. The current scatter plots do not effectively convey the contrast between the two experiments. Instead, bar plots would allow a clearer demonstration of the simulation errors for each experiment across different regions. To enhance interpretability, the regions could be categorized into Marine, Urban, and Mountain environments, using distinct colors for each category. Additionally, to examine

whether the improvement in CCN simulations by Inorg_Org is statistically significant, error bars should be added also.

**Response**: Thank you for the thoughtful suggestion. As recommended, we have revised Figure 1 to use bar plots instead of scatter plots to better compare the performance between the two simulations across different regions (categorized into Marine, Urban, Rural, and Mountain environments). The original scatter plot has been moved to the supplementary material as Figure S18.

We have revised the original description in **Section 3**:

**Original Paragraph:**
"CCN number concentrations in the Inorg_Org model at 0.1%, 0.2%, 0.5%, and 1% supersaturation (ss) show better agreement with measurements from various locations compared to the Inorg model (Fig. 1). The underestimation of CCN numbers in the Inorg simulation is alleviated by incorporating organic-related NPF, especially over rural and mountainous regions such as Steamboat Springs, Shouxian, and Nainital (Fig. 1), where both nucleation and initial growth rates are dominated by biogenic pathways. However, there remains a slight underestimation of CCN number in Shouxian at all supersaturation levels. This is likely due to the neglect of anthropogenic-derived HOMs to nucleation and growth, which are key NPF mechanisms in rural regions of China. The increase in CCN number due to the addition of organic NPF mechanisms is simulated not only in the locations listed in Table 1 but also on a global scale (see Fig. 7). "In urban regions of Brazil (Manacapuru), the overestimation of CCN numbers at 0.5% and 1% ss in Inorg is exacerbated (Fig. 1). Apart from urban regions like Manacapuru, Brazil, the overestimation also increases over oceanic regions such as Barrow and Graciosa. These overestimations in CCN numbers in the Inorg model are likely related to the overestimation of $H_2SO_4$ concentration in CAM6-Chem (Shao et al., 2024). Overall, the normalized mean bias (NMB) of CCN numbers at different supersaturation levels decreases from -35% (Inorg) to -24% (Inorg_Org), indicating that the Inorg_Org model provides a more accurate representation of organic contributions for further quantification in Section 3."

**Revised Paragraph:**

"The underestimation of CCN numbers in the Inorg simulation is alleviated by incorporating organic-related NPF, especially over rural and mountainous regions (Fig. 1), where both nucleation and initial growth rates are dominated by biogenic pathways. The remaining underestimation of CCN in rural regions (Fig. 1) is likely due to the neglect of anthropogenic-derived HOMs, which may play a key role in NPF in these areas. The increase in CCN number

due to the addition of organic NPF mechanisms is simulated not only in the locations listed in Table 1 but also on a global scale (see Fig. 7). In urban regions, the overestimation of CCN numbers is exacerbated (Fig. 1). These overestimations in CCN numbers in the Inorg model are likely related to the overestimation of $H_2SO_4$ concentration in CAM6-Chem (Shao et al., 2024). Overall, the relative bias of CCN numbers at different supersaturation levels decreases from -57% (Inorg) to -45% (Inorg_Org) (Fig. S18), indicating that the Inorg_Org model provides a more accurate representation of organic contributions for further quantification in Section 3."

[Figure]

**Figure 1.** Box plots showing the relative bias (%) between simulated monthly mean and observed median CCN number concentrations across categorized background sites (Marine, Urban, Mountain, Rural). Red and green boxes represent the Inorg and Inorg_Org experiments, respectively. Black triangles indicate the mean relative bias for each category. Numerical values above the boxes denote the corresponding mean normalized mean bias (NMB) for each experiment. Information on the measurement sites is provided in Table 2.

[Figure]

**Figure S18.** Comparison of simulated monthly mean and observed median CCN number concentrations (unit: $cm^{-3}$) at categorized background sites. Results from the Inorg_Org experiment are shown in green, and those from the Inorg experiment are shown in red. Information on the measurement sites is provided in Table 2. Normalized mean bias (NMB) and correlation values are indicated in the top-left corner of each panel.

**Specific comments:**

L30-32: The authors may consider removing the sentences "while the greater … nucleation rates involving sulfuric acid and organics". The current sentences look confusing here and disrupt the 2 reading flow. Alternatively, if the authors wish to retain these sentences, they need to rephrase these sentences to more clearly align with the argument proposed by Gordon et al. (2016).

**Response**: This sentence has been deleted in the revised manuscript to eliminate confusion. Also, to improve clarity and better align with the argument proposed by Gordon et al. (2016), the sentence in Lines 28-29 was modified as (The underlined content is newly added or modified):

"~~Unlike the findings of Gordon et al. (2016), the reduction is mainly driven by a greater enhancement of the sub-20 nm growth rate (GR) in the PI atmosphere compared to PD instead of the ~1 nm nucleation 30 rate ($j_{1.7nm}$).~~ The reduction is mainly driven by a greater enhancement of the sub-20 nm growth rate (GR) in the PI atmosphere compared to PD, instead of the findings of Gordon et al. (2016) that the ~1 nm nucleation rate ($j_{1.7nm}$) drives the reduction."

L57: "are they" to "they are"

L59: "stringent" to "rigorous"

L213: "simulated" to "found"

L215: "a low" to "the originally low"

L221: "in PI" to "in the PI"

**Response:** Thanks. We have made the corresponding revisions in the manuscript based on your suggestions.

L171-172: It is difficult to discern the regional variation in bias from Figure 1.

**Response:** I have made the revision to Figure 1 based on Comment #3. We hope this has improved the figure's readability.

L224: Add description for panels (c) and (d)

**Response:** Sorry for the missing information. The caption for Figure 4 has been updated to include descriptions for panels (c) and (d) (The underlined content is newly added):

[Figure]

Figure 4.  The relative change (unitless) of the simulated (a and b) vertically-integrated nucleation rate ($j_{1.7nm}$, below 15 km) and (c and d) vertically-mean sub-20nm growth rate after adding organic nucleation is shown in PD and PI environments. Global mean values are shown on the top right of each figure.

L229: CCN number concentration?

**Response:** The sentence in Line 229 was modified as (The underlined content is newly added or modified):

"The significant increase in CCN  burden (Fig. 2) and CDNC (Fig. 3) in the PI experiment resulting from the inclusion of the organic NPF scheme is likely to reduce the aerosol radiative forcing. "

L230: But adding the NPF mechanism would increase aerosol burden, and thus change the direct radiative forcing, although the size distribution might not change.

**Response**: Thank you for your suggestion. We have added the total effective aerosol forcing and effective radiative forcing due to aerosol-radiation interactions (ERF$_{ari}$) changes in a table (Table S4) to provide a clearer understanding.

**Table S4.** Decomposition of the global aerosol radiative forcing in different experiments (W m$^{-2}$).

|  | Inorg_Org (W m$^{-2}$) | Inorg (W m$^{-2}$) | Inorg_Org-Inorg (W m$^{-2}$) |
|---|---|---|---|
| Total effective aerosol forcing | -2.19 | -2.64 | 0.45 |
| Effective radiative forcing due to aerosol-cloud interactions (ERF$_{aci}$) | -2.18 | -2.59 | 0.41 |

| Effective radiative forcing due to aerosol-radiation interactions (ERF$_{ari}$) | 0.03 | -0.01 | 0.04 |
| --- | --- | --- | --- |

The description of the aerosol forcing in Line 230 of Section 4.2 was modified as (The underlined content is newly added or modified):

"The significant increase in CCN number and CDNC in the PI experiment resulting from the inclusion of the organic NPF scheme is likely to reduce the aerosol radiative forcing.   In this study we  focus on  quantifying the effect of including biogenic organic NPF on the indirect aerosol forcing component (ERF$_{aci}$)."

The description of the aerosol forcing in Line 250 of Section 4.2 was modified as (The underlined content is newly added or modified):

" We estimate that the global ERF$_{aci}$ since 1850, after including organic NPF, is -2.18 W m$^{-2}$ (Fig. 6a). The calculated aerosol ERF$_{aci}$ decreases by approximately 0.4 W m$^{-2}$ (corresponding to a 16% reduction) after adding organic NPF mechanisms (Fig. 6b). The global mean effective radiative forcing due to aerosol-radiation interactions (ERF$_{ari}$) changes only slightly, from 0.03 W m$^{-2}$ to -0.01 W m$^{-2}$, a negligible change compared to the total aerosol radiative forcing, which decreases from -2.19 W m$^{-2}$ to -2.64 W m$^{-2}$ (Table S4)."

L278: "in PI" to "in the PI"

L278: "leading" to "leads"

L281: "compared to PI" to "compared to the PI"

L284: "in PI" to "in the PI"

L295: remove "was neglected"

**Response:** Thank you for your suggestions. We have made these revisions.

L312: Be specific about backgrounds

**Response:** The sentence in Lines 312 was modified as (The underlined content is newly added or modified):

"After incorporating organic NPF scheme with state-of-the-art chemical mechanisms for biogenic HOMs into CAM6-Chem, the simulated CCN numbers agree better with

measurements across different backgrounds (including mountain, rural, and marine) (Fig. 1)."

L315-320: To be clearer, the authors should first clarify how organic nucleation changes are responsible for greater enhancement in PI's CCN burden in previous studies or Gordon et al. (2016). They can then highlight how their findings differ from those earlier results.

**Response**: Thank you for the suggestion. The sentence in Lines 315 was modified as (The underlined content is newly added or modified):

"After incorporating organic NPF scheme with state-of-the-art chemical mechanisms for biogenic HOMs into CAM6-Chem, the simulated CCN numbers agree better with measurements across different backgrounds (Fig. 1). Globally, the inclusion of organic-related NPF processes results in a 39% increase in CCN burden in the PI experiment and an 18% increase in the PD experiment. Similarly, cloud droplet number concentration (CDNC) at the top of low clouds in the Inorg_Org simulation rises by 12% in the PI experiment but only by 7% in the PD experiment. The greater enhancement of CCN burden in the PI experiment is primarily driven by organic condensational growth on sub-20 nm particles, rather than organic nucleation. We noted that previous studies (Zhu et al., 2019; Gordon et al., 2016) attributed the greater enhancement of CCN burden in the PI experiment to organic nucleation, which is likely due to an overestimation of the organic nucleation rate by assuming uniform volatility among all organic nucleating species. In our study, only accretion products generated through self- and cross-reactions of biogenic radicals are allowed to contribute to pure organic nucleation, making heteromolecular nucleation ($J_{SA-Org}$) the dominant nucleation pathway. Higher $H_2SO_4$ concentrations in the PD environment further enhance nucleation rates compared to the PI atmosphere.  In contrast, HOM concentrations are higher in the PI atmosphere (Fig. S3) leading to a much greater condensation of organics on sub-20 nm particles in PI experiments."

L320: add a period at the end of the sentence.

L336: remove "the"

L340: remove the extra period

**Response**: Thank you for your suggestions. The corresponding revisions have been made in the manuscript.

L328-330: Rephase the sentences to emphasize "although our methods improve the simulations of CCN burdens"

**Response**: The sentence in Lines 328 was modified as (The underlined content is newly added or modified):

"Although we improve the simulations of CCN numbers by utilizing utilized explicit chemical reactions to replace the traditional fixed yield method for simulating biogenic HOMs concentrations, further studies are needed to better align simulated HOM concentrations with widespread measurements."

**Reference**

Gordon, H., Sengupta, K., Rap, A., Duplissy, J., Frege, C., Williamson, C., Heinritzi, M., Simon, M., Yan, C., Almeida, J., Trostl, J., Nieminen, T., Ortega, I. K., Wagner, R., Dunne, E. M., Adamov, A., Amorim, A., Bernhammer, A. K., Bianchi, F., Breitenlechner, M., Brilke, S., Chen, X. M., Craven, J. S., Dias, A., Ehrhart, S., Fischer, L., Flagan, R. C., Franchin, A., Fuchs, C., Guida, R., Hakala, J., Hoyle, C. R., Jokinen, T., Junninen, H., Kangasluoma, J., Kim, J., Kirkby, J., Krapf, M., Kurten, A., Laaksonen, A., Lehtipalo, K., Makhmutov, V., Mathot, S., Molteni, U., Monks, S. A., Onnela, A., Perakyla, O., Piel, F., Petaja, T., Praplanh, A. P., Pringle, K. J., Richards, N. A. D., Rissanen, M. P., Rondo, L., Sarnela, N., Schobesberger, S., Scott, C. E., Seinfeldo, J. H., Sharma, S., Sipila, M., Steiner, G., Stozhkov, Y., Stratmann, F., Tome, A., Virtanen, A., Vogel, A. L., Wagner, A. C., Wagner, P. E., Weingartner, E., Wimmer, D., Winkler, P. M., Ye, P. L., Zhang, X., Hansel, A., Dommen, J., Donahue, N. M., Worsnop, D. R., Baltensperger, U., Kulmala, M., Curtius, J., and Carslaw, K. S.: Reduced anthropogenic aerosol radiative forcing caused by biogenic new particle formation, P. Natl. Acad. Sci. USA, 113, 12053-12058, 10.1073/pnas.1602360113, 2016.

Zhu, J., Penner, J. E., Yu, F., Sillman, S., Andreae, M. O., and Coe, H.: Decrease in radiative forcing by organic aerosol nucleation, climate, and land use change, Nat. Commun., 10, 423, 10.1038/s41467-019-08407-7, 2019.

---

## Author Comment (AC3)

We are very grateful to the evaluations from the reviewers, which have allowed us to clarify and improve the manuscript. Below we addressed the reviewer comments, with the reviewer comments in **black** and our response in **blue**.

**Reply for the referee comment#3**

**General Comments:**

The manuscript addresses a significant knowledge gap regarding the role of highly oxygenated organic molecules (HOMs) in new particle formation (NPF) during the preindustrial era, which is critical for establishing accurate baseline conditions for radiative forcing calculations. The implementation of a semi-explicit HOMs chemistry scheme (in CAM6-Chem) represents a significant improvement compared to the simplified fixed-yield approaches. A very interesting finding is that the condensational growth (before Aitken mode particles are formed) rather than nucleation itself is the primary driver of CCN enhancement in preindustrial conditions. The manuscript is well written, with great supporting materials that help the readers to better understand the results. On the other hand, I think the manuscript could be further improved in the following aspects:

**Response**: We thank the referee for the positive and insightful comments, which indeed help us further improve the manuscript. We have incorporated the necessary revisions to address the comments, and the corresponding line numbers have been indicated in the revised manuscript. Please see the revision and the response for the comments as follows.

**Comment#1:** The manuscript identifies enhanced growth rates of small particles in the PI atmosphere as the primary driver for increased CCN concentrations and subsequent reduction in ERFaci. Since the impact of new treatment on sub-20nm particle growth under preindustrial (PI) conditions is one of the key findings of this study, it would be beneficial to include more analysis and discussion on this topic in the main text. For example, it would be useful to present figure S13 (panels c,d) and corresponding results for "PI_Inorg" and compare the growth rates between the two cases under PD&PI conditions. If there are model diagnostics of condensation rates by sulfuric acid gas and by organics, it would be valuable to compare them.

**Response**: Thank you for this valuable suggestion. We have revised the main text to include further discussion on this topic. Specifically, we have moved Figure S13 panels (c, d) into the main manuscript (now shown in Fig. 8) and added the growth rate in "PI_Inorg" to compare the growth rates between the two cases under both PD and PI conditions. Additionally, the condensation rates of $H_2SO_4$ gas and organics in PD and PI experiments are now added in the supplementary materials (Fig. S14).

Sentences in Lines 283-285 were modified as (The underlined content is newly added):

"Therefore, the greater enhancement of CCN burden in the PI experiment and the reduction in ERF$_{aci}$ are mainly caused by organic condensational growth on sub-20 nm particles (with PD fractional changes of 6% in Inorg_Org and 58% in Inorg; Fig. 7), rather than by organic nucleation. Specifically, after incorporating the organic NPF mechanism, the growth rate of sub-20 nm particles increases more significantly in the PI experiment (0.0083 nm h$^{-1}$) than in the PD experiment (0.0036 nm h$^{-1}$) (Fig. 8). This is mainly due to the higher organic sub-20 nm growth rate in PI (0.01 nm h$^{-1}$) compared to PD (0.006 nm h$^{-1}$)."

[Figure]

**Figure 8.** Spatial distribution of the simulated vertically-mean growth rate in (a and d) PD and (b and e) PI experiments. The difference between Inorg_Org and Inorg in PD and PI experiments is shown in (c) and (f) (unit: nm h$^{-1}$). Global mean values are shown on the top right of each figure.

[Figure]

**Figure S14.** Spatial distribution of the simulated vertically-mean inorganic growth rate (a and b) and organic growth rate (c and d) in PD and PI experiments (unit: nm h$^{-1}$). Global mean values are shown on the top right of each figure.

**Comment#2:** The present study discusses the impact of the new nucleation treatment on indirect aerosol effect. It would be more meaningful to also present total effective aerosol forcing changes (either in figure or table), in addition to the decomposed values, since there are often compensating effects between different forcing components.

**Response:** Thank you for your suggestion. We have added the total effective aerosol forcing and effective radiative forcing due to aerosol-radiation interactions (ERF$_{ari}$) changes in a table (Table S4) to provide a clearer understanding.

The description of the aerosol forcing in Line 250 of Section 4.2 was modified as (The underlined content is newly added or modified):

"We estimate that the global ERF$_{aci}$ since 1850, after including organic NPF, is -2.18 W m$^{-2}$ (Fig. 6a). The calculated aerosol ERF$_{aci}$ decreases by approximately 0.4 W m$^{-2}$ (corresponding to a 16% reduction) after adding organic NPF mechanisms (Fig. 6b). The global mean effective radiative forcing due to aerosol-radiation interactions (ERF$_{ari}$) changes only slightly, from 0.03 W m$^{-2}$ to -0.01 W m$^{-2}$, a negligible change compared to the total aerosol radiative forcing, which decreases from -2.19 W m$^{-2}$ to -2.64 W m$^{-2}$ (Table S4)."

**Table S4.** Decomposition of the global aerosol radiative forcing in different experiments (W m$^{-2}$).

| | Inorg_Org (W m$^{-2}$) | Inorg (W m$^{-2}$) | Inorg_Org-Inorg (W m$^{-2}$) |
|---|---|---|---|
| Total effective aerosol forcing | -2.19 | -2.64 | 0.45 |
| Effective radiative forcing due to aerosol-cloud interactions (ERF$_{aci}$) | -2.18 | -2.59 | 0.41 |
| Effective radiative forcing due to aerosol-radiation interactions (ERF$_{ari}$) | 0.03 | -0.01 | 0.04 |

**Comment#3:** Since the authors claim (line 284 and 316) that the CCN burden change and the indirect aerosol effect are mainly caused by ORGANIC condensational growth on sub-20nm particles, it would be useful to isolate this impact by performing an additional simulation with the organic condensational growth switched off (if it is straightforward). Otherwise, I would suggest stating it as "likely" or removing the emphasis on organics condensation.

**Response**: We acknowledge that this diagnostic quantity was not initially provided. Due to the significant computational resources required for long-term simulations, we are unable to run an additional set of experiments with organic condensational growth switched off.

We believe that Figure 4 in the main text provides strong support for this conclusion. In Figure 4, the impact of organic nucleation on the total nucleation rate in the PI atmosphere is relatively small, accounting for only 6%. This is mainly due to the lower H$_2$SO$_4$ concentrations in the PI atmosphere (Fig. S2), which reduces the H$_2$SO$_4$-HOM nucleation rate. In contrast, in the PD atmosphere, higher concentrations of H$_2$SO$_4$ and HOMs in the Northern Hemisphere mid-to-high latitudes (Figs. S2 and S3) make organic nucleation more significant, leading to a 39% increase in the total nucleation rate. In contrast to organic nucleation, the impact of organic growth rate on the total growth rate is more significant in the PI atmosphere, reaching 83%, while in the PD atmosphere this impact is only 23%. This is mainly due to significantly higher emissions of organic precursors, such as monoterpenes and isoprene in the PI atmosphere (Fig. S3), and the organic growth rate is only influenced by HOMs concentrations. Therefore, when considering the greater increase in CCN burden in the PI experiment compared to the PD experiment (Fig. 2), the increase in sub-20 nm growth rate plays a more significant role than nucleation rate (Fig. 4 and Fig. S13).

To further support this conclusion, we have calculated the correlation between condensational growth rate on sub-20nm particles with CCN burden. A higher correlation (R ~ 0.7) between organic condensational growth rate and CCN burden, as well as a relatively lower correlation (R ~ 0.4), further suggests that the increase in CCN burden is likely driven by organic

condensational growth. Additionally, the Figure 8 added in Response 1 may provide further support for this point.

[Figure]

Figure 4. The relative change (unitless) of the simulated (a and b) vertically-integrated nucleation rate ($J_{1.7nm}$, below 15 km) and (c and d) vertically-mean sub-20nm growth rate after adding organic nucleation is shown in PD and PI environments. Global mean values are shown on the top right of each figure.

[Figure]

**Figure S15.** Global mean values of simulated CCN burden at 0.2% supersaturation (blue lines, unit: $10^6$ cm$^{-2}$) and vertically averaged (a) organic and (b) inorganic growth rates (unit: nm h$^{-1}$) in the

PD_Inorg_Org experiment. Pearson correlation coefficients between growth rates and CCN burden are shown in the titles of each panel.

The description of the aerosol forcing in Line 213 of Section 4.1 were modified as (The underlined content is newly added or modified):

"In both PD and PI experiments, the largest increase in CCN burden (>20% rise in Inorg_Org compared to Inorg) is simulated in the tropical regions (Amazon, central Africa, and Southeast Asia) (Fig. 2). This is attributed to the highest biogenic emissions (Fig. S3) which lead to the greatest increases in both nucleation and growth rates in Inorg_Org (Fig. 4) and the originally low aerosol number before adding organic NPF (i.e., Inorg simulation) in these regions. The enhancement in nucleation rates due to the inclusion of organic nucleation is more significant in the PD experiment (39%) compared to the PI experiment (6%) (Fig. 4). This is mainly caused by higher sulfuric acid concentrations in PD environment (Fig. S2), resulting in higher heteromolecular nucleation rates involving sulfuric acid and organics (Figs. S6 and S7). A detailed discussion of the specific reason is provided in Section 4.2. In contrast to organic nucleation, the impact of organic growth rate on the total growth rate is more significant in the PI atmosphere, reaching 83%, while in the PD atmosphere this impact is only 23%. This is mainly due to significantly higher emissions of organic precursors, such as monoterpenes and isoprene in the PI atmosphere (Fig. S3), and the organic growth rate is only influenced by HOMs concentrations. Therefore, comparing to the increase in the ~1.7 nm nucleation rate, the increase in the sub-20 nm growth rate plays a more significant role in greater increase of CCN burden in PI experiment (Fig. 4 and Fig. S13). The strong correlation (R~0.7) between organic growth rates and CCN burden further supports this point (Fig. S15)."

The description of the aerosol forcing in Line 283 was modified as (The underlined content is newly added or modified):

"Therefore, the greater enhancement of CCN burden in PI experiment and reduction in ERF$_{aci}$ are mainly likely caused by organic condensational growth on sub-20nm particles instead of organic nucleation."

**Specific comments:**
Figure 3: Is CDNC either vertically-integrated or at the top of low clouds? Which is correct?

**Response**: CDNC is calculated at the cloud top of low-level clouds. We will update the figure caption to state this explicitly. Figure 3 will be revised as:

[Figure]

**Figure 3.** Spatial distribution of the simulated  cloud droplet number concentration (CDNC) at the top of low clouds in (a) PD_Inorg_Org and (b) PI_Inorg_Org (unit: $cm^{-3}$). The relative change after adding organic NPF  in PD and PI environments are shown in (c) and (d). Global mean values are shown on the top right of each figure.

Figure 4: Why are the values over ocean negative in panels a) and b)? Also, the caption is somewhat confusing. I suggest rewriting it.

**Response**: The main reason for the negative values over the ocean in panels a) and b) is that, after including organic NPF, heteromolecular nucleation involving $H_2SO_4$ and organics ($J_{SA-Org}$) consumes more $H_2SO_4$ over land, as monoterpene emissions and the formation of HOMs are mainly concentrated over there (Fig. S3). As a result, less $H_2SO_4$ is transported to the ocean, where nucleation (Figs. S4 and S5) and sub-20 nm particle growth are mainly driven by $H_2SO_4$ (Fig. 7 in Shao et al., 2024), leading to decreased nucleation rate and $H_2SO_4$ nucleation sink (Fig. S16) over ocean.

In light of this, we have added the following explanation in Line 219 of the main text (The underlined content is newly added or modified):

"The enhancement in nucleation rates due to the inclusion of organic nucleation is more significant in the PD experiment (39%) compared to the PI experiment (6%) (Fig. 4). This is mainly caused by higher sulfuric acid concentrations in PD environment (Fig. S2), resulting in higher heteromolecular nucleation rates involving sulfuric acid and organics ($J_{SA-Org}$) (Figs. S6 and S7) over land, where both $H_2SO_4$ and HOMs show high value. Consequently, more $H_2SO_4$ is consumed over land (Fig. S16), reducing its transport to oceanic regions. As a result, nucleation rates decrease over the ocean in both the PD and PI experiments (Fig. 4)."

[Figure]

**Figure S16.** Spatial distribution of the simulated vertically-integrated sulfuric acid nucleation loss rate in (a) PD and (b) PI experiments (unit: ng m$^{-2}$ s$^{-1}$). The relative difference between Inorg_Org and Inorg in PD and PI experiments are shown in (c) and (d) (unitless). Global mean values are shown on the top right of each figure.

The original caption of Figure 4 was revised as below:

[Figure]

Figure 4.  The relative change (unitless) of the simulated (a and b) vertically-integrated nucleation rate ($j_{1.7nm}$, below 15 km) and (c and d) vertically-mean sub-20nm growth rate after adding organic nucleation is shown in PD and PI environments. Global mean values are shown on the top right of each figure.

Figure 6: It would be more meaningful to also present total effective aerosol forcing changes (either in figure or table), in addition to the decomposed values. There are often compensating effects between different forcing components.

**Response**: Thank you for your suggestion. In response to your second major comment, we have made the necessary revisions and included the total effective aerosol forcing changes, in addition to the decomposed values, as recommended.

Line 176: Should be Table 2.

Line 198: "rise" → "rises"

Line 220: "comparing to" → "compared to"

**Response**: Thank you for your helpful comments. In response to your suggestions, we have made the following revisions in the manuscript:

Line 176: Corrected to "Table 2" as suggested. Line 198: Changed "rise" to "rises". Line 220: Replaced "comparing to" with "compared to".

Line 230-231: Is this really the case in this study? It appears that the accumulation mode aerosol number concentrations have significant changes after considering organic nucleation.

**Response**: Thank you for your suggestion. We have removed the sentence and added the total effective aerosol forcing and effective radiative forcing due to aerosol-radiation interactions (ERF$_{ari}$) changes in a table (Table S4) to provide a clearer understanding.

**Table S4.** Decomposition of the global aerosol radiative forcing in different experiments (W m$^{-2}$).

| | Inorg_Org (W m$^{-2}$) | Inorg (W m$^{-2}$) | Inorg_Org-Inorg (W m$^{-2}$) |
|---|---|---|---|
| Total effective aerosol forcing | -2.19 | -2.64 | 0.45 |
| Effective radiative forcing due to aerosol-cloud interactions (ERF$_{aci}$) | -2.18 | -2.59 | 0.41 |
| Effective radiative forcing due to aerosol-radiation interactions (ERF$_{ari}$) | 0.03 | -0.01 | 0.04 |

The description of the aerosol forcing in Line 230 of Section 4.2 were modified as (The underlined content is newly added or modified):

"The significant increase in CCN number and CDNC in the PI experiment resulting from the inclusion of the organic NPF scheme is likely to reduce the aerosol radiative forcing.  In this study we  focus on quantifying the effect of including biogenic organic NPF on the indirect aerosol forcing component (ERF$_{aci}$)."

The description of the aerosol forcing in Line 250 of Section 4.2 were modified as (The underlined content is newly added or modified):

" We estimate that the global ERF$_{aci}$ since 1850, after including organic NPF, is -2.18 W m$^{-2}$ (Fig. 6a). The calculated aerosol ERF$_{aci}$ decreases by approximately 0.4 W m$^{-2}$ (corresponding to a 16% reduction) after adding organic NPF mechanisms (Fig. 6b). The global mean effective radiative forcing due to aerosol-radiation interactions (ERF$_{ari}$) changes only slightly, from 0.03 W m$^{-2}$ to -0.01 W m$^{-2}$, a negligible change compared to the total aerosol radiative forcing, which decreases from -2.19 W m$^{-2}$ to -2.64 W m$^{-2}$ (Table S4)."

Line 280: "organics rate" → "organics"

Line 295: Delete "was neglected"

Line 318: "is" → "are"

Line 324: Should be Figure 6 (not 9).

Line 325: "higher increase" → "greater increase"

**Response**: Thanks. We have made the corresponding revisions in the manuscript based on your suggestions.

Line 337: "with a greater reduction than estimated in previous studies." This appears inconsistent with Gordon et al.'s higher percentage reduction (27%) than this study (16%).

**Response**: Thank you for pointing out the inconsistency. What we intended to emphasize was the comparison with Zhu et al. (2019) (16% reduction), as Zhu et al. (2019) employed a more advanced chemical mechanism compared to Gordon et al. (2016). To avoid any confusion, we will remove this paragraph.

**Reference**

Gordon, H., Sengupta, K., Rap, A., Duplissy, J., Frege, C., Williamson, C., Heinritzi, M., Simon, M., Yan, C., Almeida, J., Trostl, J., Nieminen, T., Ortega, I. K., Wagner, R., Dunne, E. M., Adamov, A., Amorim, A., Bernhammer, A. K., Bianchi, F., Breitenlechner, M., Brilke, S., Chen, X. M., Craven, J. S., Dias, A., Ehrhart, S., Fischer, L., Flagan, R. C., Franchin, A., Fuchs, C., Guida, R., Hakala, J., Hoyle, C. R., Jokinen, T., Junninen, H., Kangasluoma, J., Kim, J., Kirkby, J., Krapf, M., Kurten, A., Laaksonen, A., Lehtipalo, K., Makhmutov, V., Mathot, S., Molteni, U., Monks, S. A., Onnela, A., Perakyla, O., Piel, F., Petaja, T., Praplanh, A. P., Pringle, K. J., Richards, N. A. D., Rissanen, M. P., Rondo, L., Sarnela, N., Schobesberger, S., Scott, C. E., Seinfeldo, J. H., Sharma, S., Sipila, M., Steiner, G., Stozhkov, Y., Stratmann, F., Tome, A., Virtanen, A., Vogel, A. L., Wagner, A. C., Wagner, P. E., Weingartner, E., Wimmer, D., Winkler, P. M., Ye, P. L., Zhang, X., Hansel, A., Dommen, J., Donahue, N. M., Worsnop, D. R., Baltensperger, U., Kulmala, M., Curtius, J., and Carslaw, K. S.: Reduced anthropogenic aerosol radiative forcing caused by biogenic new particle formation, P. Natl. Acad. Sci. USA, 113, 12053-12058, 10.1073/pnas.1602360113, 2016.

Zhu, J., Penner, J. E., Yu, F., Sillman, S., Andreae, M. O., and Coe, H.: Decrease in radiative forcing by organic aerosol nucleation, climate, and land use change, Nat. Commun., 10, 423, 10.1038/s41467-019-08407-7, 2019.